# Prostaglandin F2α regulates mitochondrial dynamics and mitophagy in the bovine corpus luteum

Michele R Plewes[1,2,3], Emilia Przygrodzka[1], Corrine F Monaco[1,4], Alexandria P Snider[5], Jessica A Keane[5], Patrick D Burns[6], Jennifer R Wood[5], Andrea S Cupp[5], John S Davis[1,2,3]

**Prostaglandins are arachidonic acid-derived lipid mediators involved in numerous physiological and pathological processes. PGF2α analogues are therapeutically used for regulating mammalian reproductive cycles and blood pressure, inducing term labor, and treating ocular disorders. PGF2α exerts effects via activation of calcium and PKC signaling, however, little is known about the cellular events imposed by PGF2α signaling. Here, we explored the early effects of PGF2α on mitochondrial dynamics and mitophagy in the bovine corpus luteum employing relevant and well characterized in vivo and in vitro approaches. We identified PKC/ERK and AMPK as critical protein kinases essential for activation of mitochondrial fission proteins, DRP1 and MFF. Furthermore, we report that PGF2α elicits increased intracellular reactive oxygen species and promotes receptor-mediated activation of PINK–Parkin mitophagy. These findings place the mitochondrium as a novel target in response to luteolytic mediator, PGF2α. Understanding intracellular processes occurring during early luteolysis may serve as a target for improving fertility.**

## Introduction

The corpus luteum is a transient endocrine gland that secretes the steroid hormone progesterone to maintain pregnancy (1, 2). This ephemeral gland is inversely regulated by luteotrophic hormones which support luteal formation, maintenance, and steroidogenesis (3), and luteolytic hormones, such as prostaglandin F2α (PGF2α), which trigger loss of progesterone and regression of the gland (1). Luteolysis is a naturally occurring event necessary for regulation of the female reproductive cycle (4). At the onset of luteolysis, there is a precipitous decline in serum progesterone concentrations

(functional regression) followed by loss of luteal weight (structural regression) (5). PGF2α released from the uterus is responsible for initiating timely luteolysis in several non-primate species (6, 7, 8, 9, 10, 11, 12, 13) and primates (both endogenous PGF2α and estrogen) (14). PGF2α exerts its actions through receptor-mediated stimulation of the phospholipase C-intracellular calcium-PKC pathway and activation of downstream protein kinases, such as extracellular regulated protein kinase (ERK) (15), calcium/calmodulin-dependent protein kinase II (CaMKII) (16), and 5′-adenosine monophosphate-activated protein kinase (AMPK) (17). Although PGF2α analogues are used extensively to terminate the reproductive cycle to synchronize the ovulatory process (18), little is known about the intracellular events imposed by PGF2α receptor (PTGFR) signaling.

Mitochondria are central to many cellular physiological processes that control tissue homeostasis, including cell fate, differentiation, proliferation, and cell death (19, 20). Mitochondrial fission is a cellular mechanism that synchronously controls mitochondrial quality. Dynamin-like 1 protein (DNM1L; commonly referred to as DRP1) is a key mitochondrial GTPase responsible for controlling mitochondrial fission and is a major contributor to the manifestation and pathogenesis of various diseases (21, 22, 23, 24). DRP1 is a cytoplasmic protein that has a C-terminal GTPase effector domain, a small variable domain, a dynamin-like middle assembly domain, and an N-terminal GTP-binding domain thought to provide the mechanical force required for mitochondrial division (25). DRP1 is differentially regulated by posttranslational modifications that govern translocation to mitochondria and induction of mitochondrial fission (26). Phosphorylation of DRP1 within the GTPase effector domain at Ser637 inhibits DRP1 GTPase activity, whereby promoting mitochondrial elongation (27). In contrast, phosphorylation of DRP1 at Ser616 is mediated by PKC (28) and ERK (29) signaling. Phosphorylation of DRP1 at residue Ser616 does not directly affect GTPase activity (30), but rather mediates recruitment of DRP1 to

[1]Olson Center for Women's Health, Department of Obstetrics and Gynecology, University of Nebraska Medical Center, Nebraska Medical Center, Omaha, NE, USA [2]Department of Biochemistry and Molecular Biology, University of Nebraska Medical Center, Nebraska Medical Center, Omaha, NE, USA [3]U.S Department of Veterans Affairs Nebraska Western Iowa Health Care System, Omaha, NE, USA [4]Department of Cellular and Integrative Physiology, University of Nebraska Medical Center, Nebraska Medical Center, Omaha, NE, USA [5]Department of Animal Sciences, University of Nebraska–Lincoln, Lincoln, NE, USA [6]Department of Biological Sciences, University of Northern Colorado, Greeley, CO, USA

Correspondence: michele.plewes@unmc.edu; jsdavis@unmc.edu
Alexandria P Snider's present address is USDA, Agricultural Research Service, U.S. Meat Animal Research Center, Clay Center, NE, USA

mitochondrial fission factor (MFF), the DRP1 receptor located on the outer mitochondrial membrane (31). To promote mitochondrial fission DRP1 is recruited to MFF by AMPK-induced phosphorylation of MFF at Ser146 (32).

Mitophagy is a process that selectively sequesters damaged or depolarized mitochondria into double-membrane autophagosomes for subsequent lysosomal degradation. PTEN-induced kinase 1 (PINK1) is a protein kinase that works cooperatively with the E3 ubiquitin ligase, Parkin, to monitor the mitochondrial state and tag damaged mitochondria for degradation (33). In healthy cells, mitochondria maintain a membrane potential that can be used to import PINK1 continuously into the inner mitochondrial membrane (34). Once imported inside, PINK1 is proteolytically cleaved by mitochondrial-processing peptidase and presenilin-associated rhomboid-like and immediately cleared from the outer membrane (34). In the presence of unhealthy mitochondria, PINK1 rapidly accumulates on the outer mitochondrial membrane and is activated by autophosphorylation at Ser228 (35). Activated PINK1 then phosphorylates ubiquitin at Ser65, which competes with an autoinhibitory domain within Parkin and stabilizes it in an active conformation resulting in recruitment of Parkin to the outer mitochondrial membrane (36). Once at the mitochondria, active PINK1 phosphorylates Parkin at Ser65, leading to activation and ubiquitination of molecules on the outer mitochondrial membrane. Autophagy receptors and machinery are recruited, initiating engulfment of the ubiquitinated mitochondria in LC3-positive autophagosome. The autophagosome then fuses with the lysosome allowing for the degradation of damaged mitochondria. The quality control of mitochondria has demonstrated importance in the survival and function of cells in various disease states (37), and disruption of the mitochondrial function could be an early event involved in luteolysis.

Proper control of the life span of the corpus luteum is essential for the establishment and maintenance of pregnancy in mammals (38). In the present study, we set out to delineate the effects of the luteolytic hormone, PGF2α, on mitochondrial dynamics (activation of mitochondrial fission) and mitophagy in the bovine corpus luteum. Using in vivo and in vitro approaches, we provide the initial evidence that PGF2α, via cross-communication between PKC/ERK and AMPK, regulates the phosphorylation of both DRP1 and its mitochondrial receptor, MFF, in large luteal cells leading to fission of the mitochondria. Furthermore, we demonstrate that PGF2α-mediated activation of DRP1 and MFF is accompanied by increased mitochondrial fission, reactive oxygen species (ROS) production, and activation of mitophagy. Hormonal regulation of mitochondria dynamics may be an early step for regulating luteal function at the time of luteolysis.

# Results

### Temporal effects of PGF2α on progesterone biosynthesis in vivo

To evaluate the early temporal effects of PGF2α on progesterone production, cows were administered a single dose of saline or PGF2α (i.m.) and corpora lutea were collected at zero time, 1, 2, and 4 h posttreatment. Serum progesterone was decreased 2 h post-injection of PGF2α ($P < 0.01$; Fig 1A). Moreover, there was a 50% decrease in tissue progesterone 4 h post-PGF2α treatment ($P < 0.05$; Fig 1B), independent of change in corpus luteum weight ($P > 0.05$; Fig S1A). We further evaluated the effects of PGF2α on the expression of key steroidogenic enzymes that are required for progesterone production (Fig S1B and C). We observed no difference in the content of steroidogenic enzymes STAR, CYP11A1 or HSD3B 4 h post i.m administration of PGF2α ($P > 0.05$; Fig S1D).

### Effects of PGF2α on phosphorylation of DRP1 and MFF in the bovine corpus luteum in vivo

The luteolytic hormone PGF2α acts on the large steroidogenic cells of the bovine corpus luteum (17). We determined mitochondrial dynamics, specifically the phosphorylation status of DRP1 and MFF in vivo, after administration of a single dose of saline or PGF2α (i.m.). Western blot revealed an acute 6.8-fold increase ($P$-value < 0.001) in phosphorylation of DRP1 at Ser616 1 h posttreatment with PGF2α and a 3.7- and 3.6-fold increase 2 and 4 h post-PGF2α, respectively ($P$-value < 0.01; Fig 1C and D). Western blotting revealed a 1.9-fold increase in the phosphorylation of DRP1 at Ser637 (Fig 1C and E; $P$-value < 0.05). Moreover, PGF2α stimulated a 1.7-fold increase in phosphorylation of MFF at Ser146 1 h posttreatment with PGF2α and a 1.5- and 1.4-fold increase 2 and 4 h post-PGF2α, respectively (Fig 1C and F; $P$-value < 0.05). Levels of total DRP1 and MFF protein expression were unchanged in response to PGF2α (Fig 1C; $P$-value > 0.05). Immunohistochemistry of luteal tissue revealed an observed increase in the phosphorylation of DRP1 (Ser616; Fig 1G panels a and b) and MFF (Ser146; Fig 1G panel g and h) 4 h posttreatment with PGF2α (Fig 1G). Moreover, there was a notable presence of phospho-DRP1 (Ser616) and MFF (Ser146) localized to the large luteal cell population (Fig 1G panels b and h), supporting our hypothesis that PGF2α regulates the phosphorylation of mitochondrial fission proteins, DRP1 and MFF, in luteal tissue. Negative controls are presented in Fig S1E.

### Effects of PGF2α on phosphorylation of AMPK and mitophagy machinery in the bovine corpus luteum in vivo

Next, we determined whether PGF2α induces activation of AMPK and phosphorylation of mitophagy machinery in vivo. Western blot revealed an acute 7.1-fold increase ($P$ = 0.08) in phosphorylation of AMPK at Thr172 1 h posttreatment with PGF2α and a 6.5- and 6.8-fold increase 2 and 4 h post-PGF2α, respectively ($P$-value < 0.05; Fig 2A and B). Western blotting revealed a 1.4-fold increase in the phosphorylation of PINK1 at Ser228 4 h posttreatment with PGF2α ($P$-value < 0.05; Fig 2A and C). ULK1 is an essential kinase involved in the autophagy pathway. AMPK activates ULK1 through phosphorylation at multiple sites, including serine 555, which stimulates activity for autophagy (39). Moreover, phosphorylation of ULK1 at Ser555 provides the switch from canonical autophagy to mitophagy-specific pathways after AMPK activation (40). Treatment with PGF2α stimulated a 1.4-fold increase in phosphorylation of ULK1 at Ser555 2 h posttreatment with PGF2α and a 1.6-fold increase 4 h post-PGF2α ($P$-value < 0.05; Fig 2A and D). We used

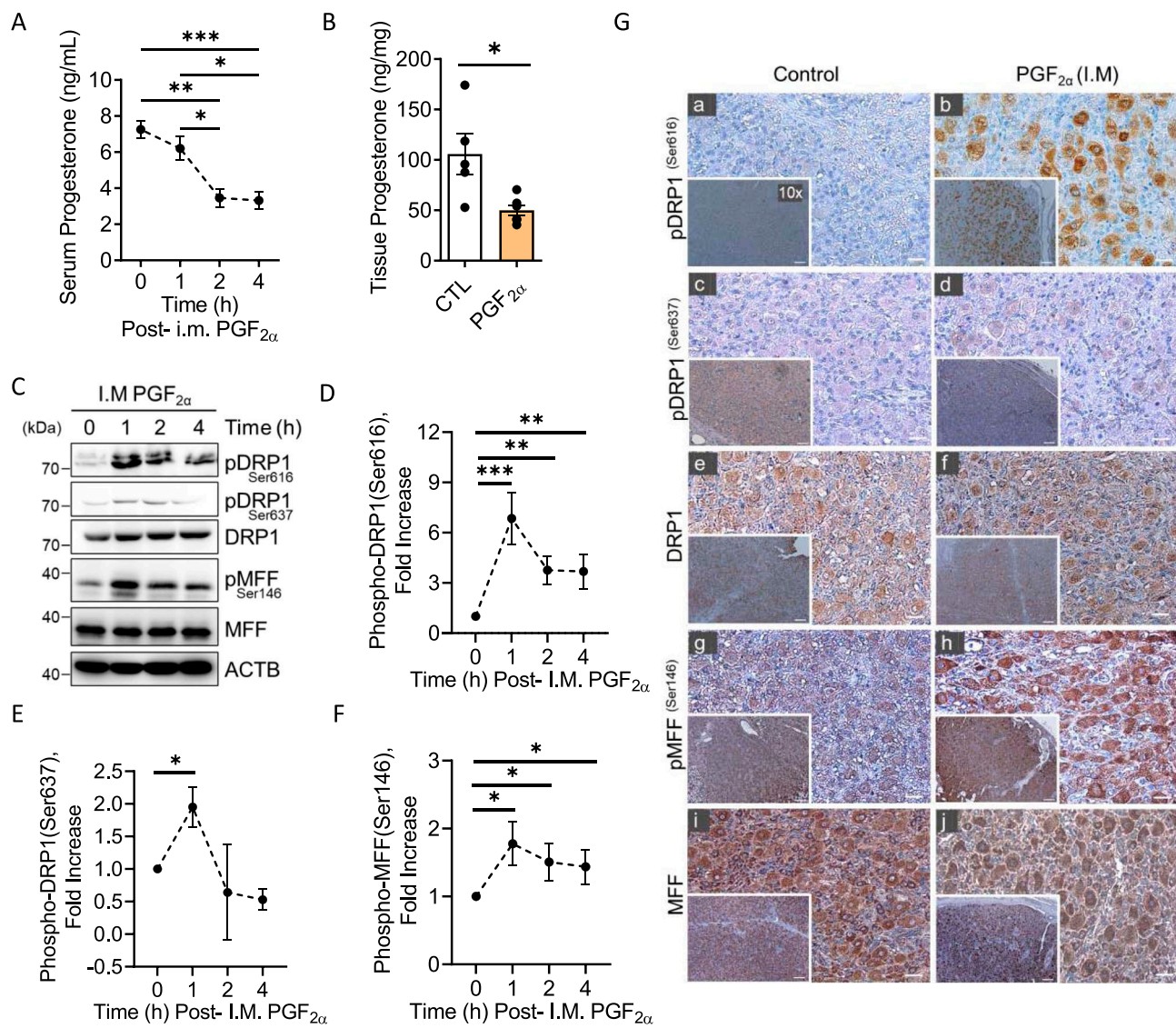

**Figure 1. Temporal effects of Prostaglandin F2alpha (PGF2α) on progesterone production and the phosphorylation of dynamin-related protein-1 (DRP1) and mitochondrial fission factor (MFF) in vivo.**
Mid-cycle cows (n = 3/time-point) were administered I.M. PGF2α (25 mg) for 1, 2, and 4 h or control saline injections (n = 3). **(A)** Serum progesterone concentrations were obtained from animals 0, 1, 2, and 4 h following I.M PGF2α administration (n = 3 to 4 animals per time point). Statistics were performed by one-way ANOVA followed by Tukey's multiple comparison tests. **(B)** Luteal tissue progesterone concentrations (n = 5 saline-treated; n = 6 PGF2α-treated) 4 h post-I.M PGF2α treatment. Statistics were performed by t tests to evaluate paired responses. **(C)** Representative Western blot analysis of the phosphorylation of DRP1 and MFF in luteal tissue 1, 2, and 4 h after I.M administration of PGF2α. **(D)** Densitometric analyses of phospho-DRP1 (Ser616). **(E)** Densitometric analyses of phospho-DRP1 (Ser637). Symbols represent mean fold changes (means ± sem, n = 3). **(F)** Densitometric analyses of phospho-MFF (Ser146). Symbols represent mean fold changes (means ± sem, n = 3). Statistics performed by two-way ANOVA was used to evaluate repeated measures with Dunnett's post tests to compare means. **(G)** Representative immunohistochemistry micrograph of the phosphorylation of DRP1 and MFF in luteal tissue 4 h after I.M administration of PGF2α treatment. Micron bar = 5 mm (10x) and 1 mm (40x). Significant difference between treatments compared with saline-treated animals, *P < 0.05; **P < 0.01; ***P < 0.001.

immunohistochemistry to determine the effects of PGF2α on the phosphorylation of AMPK and ULK1 at Ser555 in vivo (Fig 2E). We observe an increase in the phosphorylation of AMPK (Thr172; Fig 2E panels a and b) and ULK1 (Ser555; Fig 2E panels e and f), 4 h posttreatment with PGF2α. In addition, we observed an increase in the expression of phospho-PINK1 (Ser 228; Fig 2E panels c and d), and L3CB (Fig 2E panels g and h), a central protein in autophagy (41), 4 h posttreatment with PGF2α, indicative of activation of mitophagy.

### Expression of DRP1 and MFF in the bovine ovary

After ovulation, the granulosa cells of the ovarian follicle differentiate into large luteal cells. To determine the expression of DNM1L/DRP1 and MFF in the bovine ovary, we mined bovine gene expression arrays from NCBI GEO repository (GSE83524) to analyze expression of transcripts for bovine granulosa, large, and small luteal cells (Fig S2A and B) (42, 43). The DNM1L mRNA transcripts were enriched 1.8-fold in large luteal cells compared with granulosa

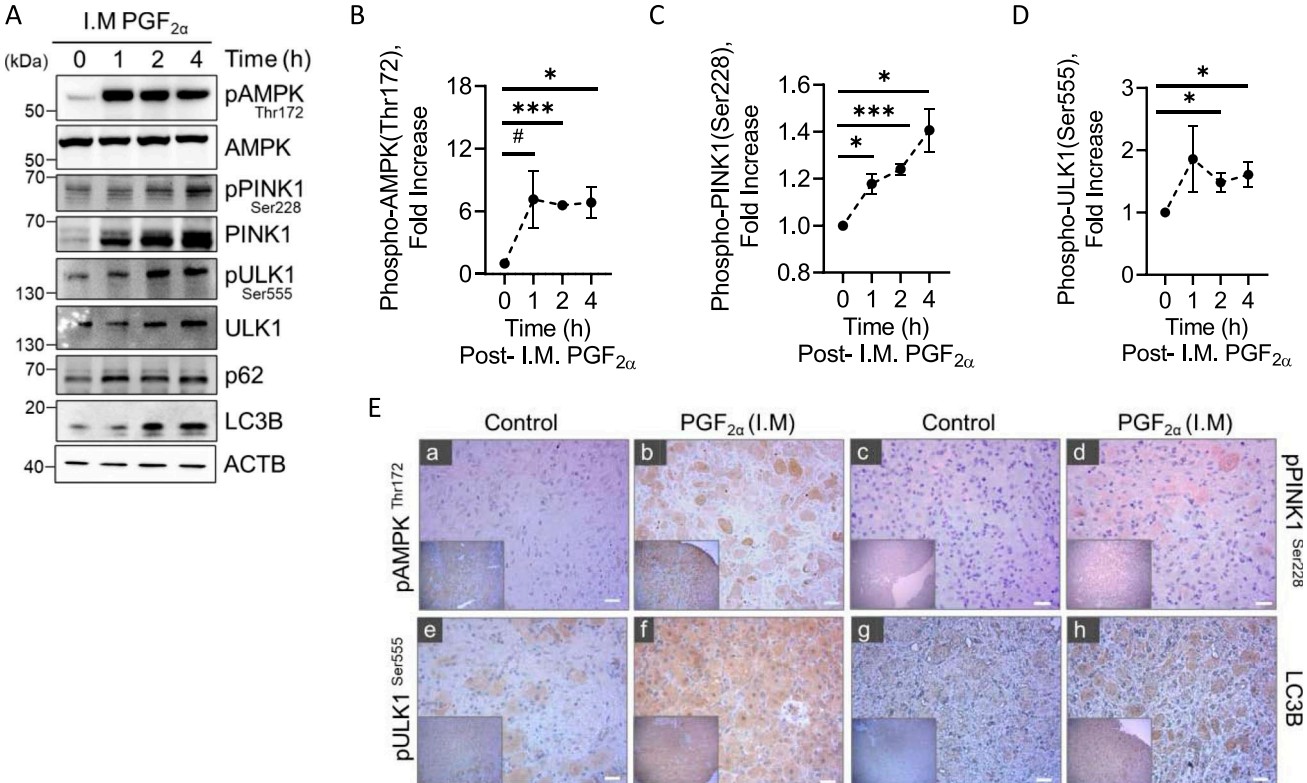

**Figure 2. Temporal effects of Prostaglandin F2alpha (PGF2α) on phosphorylation of AMP-activated protein kinase (AMPK) and Mitophagy machinery in vivo.**
Mid-cycle cows (n = 3/time-point) were administered i.m. PGF2α (25 mg) for 1, 2, and 4 h or control saline injections (n = 3). **(A)** Representative Western blot analysis of the phosphorylation of AMPK and proteins involved in the activation of mitophagy in luteal tissue 1, 2, and 4 h after i.m. administration of PGF2α. **(B)** Densitometric analyses of phospho-AMPK (Thr172). **(C)** Densitometric analyses of phospho-PINK1 (Ser228). **(D)** Densitometric analyses of phospho-ULK1 (Ser555). Symbols represent mean fold changes (means ± sem, n = 3). **(E)** Representative immunohistochemistry micrograph of the phosphorylation of AMPK1 at Thr172 (panels (a, b)), PINK1 at Ser228 (panels (c, d)), ULK1 at Ser555 (panels (e, f)), and LC3B (panels (g, h)) in luteal tissue 4 h after i.m. administration of PGF2α. Micron bar = 1 mm (40x). Statistics were performed by one-way ANOVA followed by Dunnett's post tests to compare means. Significant difference between treatments compared to saline-treated animal, #$P$ = 0.08, *$P$ < 0.05; ***$P$ < 0.001.
Source data are available for this figure.

cells ($P$ < 0.05; Fig S2A); 2.5-fold in large luteal cells compared with small luteal cells ($P$ < 0.05; Fig S2A). Transcripts for *MFF*, the DRP1 mitochondrial receptor, were not different between cell types ($P$ > 0.05; Fig S2B). By comparison, there was only a 12% difference in the expression of mRNA for *ACTB* amongst all cell types as previously reported (42, 43).

### Effects of hormones on phosphorylation of DRP1 in bovine large luteal cells

To determine the influence of luteotropic and luteolytic agents on the differential phosphorylation of DRP1, enriched populations of large luteal cells were treated for 30 min with luteotropic agents (LH or the adenylyl cyclase activator forskolin [FSK]) or luteolytic agents (PGF2α, the PKC activator phorbol ester PMA or the cytokine TNFα) (Fig S2C and D). Treatment with luteotropic hormones/activators, LH, and FSK decreased phosphorylation of DRP1 (Ser616) ($P$ < 0.05; Fig S2C and D) and stimulated phosphorylation of DRP1 at the inactivation site (Ser637) in large luteal cells (Fig S2C). When compared with control, the luteolytic hormone PGF2α increased phosphorylation of DRP1 (Ser616) 9.3-fold in large luteal cells

($P$ < 0.05, Fig S2C and D). Moreover, treatment with the PKC activator, PMA, increased phosphorylation of DRP1 (Ser616) 7.5-fold in large luteal cells when compared with control ($P$ < 0.05; Fig S2C and D). Treatment with TNFα, a cytokine involved in luteolysis (44), increased phosphorylation of DRP1 (Ser616) 2.1-fold when compared with control ($P$ < 0.05; Fig S2C and D).

Because our in vivo results showed that the localization of phospho-DRP1 (Ser616) appears to ensue in large luteal cells, enriched populations of large luteal cells were prepared and treated with luteolytic hormones, PGF2α (100 nM; Fig 3A) or TNFα (10 ng/ml; Fig S3) for up to 6 h to determine the temporal nature of the phosphorylation of DRP1. Western blot revealed that PGF2α acutely increases the phosphorylation of DPR1 (Ser616; 10.8-fold) within 30 min of treatment and phospho-DRP1 remained elevated for at least 6 h posttreatment when compared with control ($P$ < 0.05; Fig 3A and B). In addition, PGF2α increased phosphorylation of DPR1 (Ser637; 2.5-fold) 30 min posttreatment and remained elevated for 6 h posttreatment compared with control ($P$ < 0.05; Fig 3A and B). Consistent with Western blot results, PGF2α increased the mean fluorescent intensity of phospho-DRP1 (Ser616; 6.8-fold) when compared with control cells ($P$ < 0.05; Fig 3C and D).

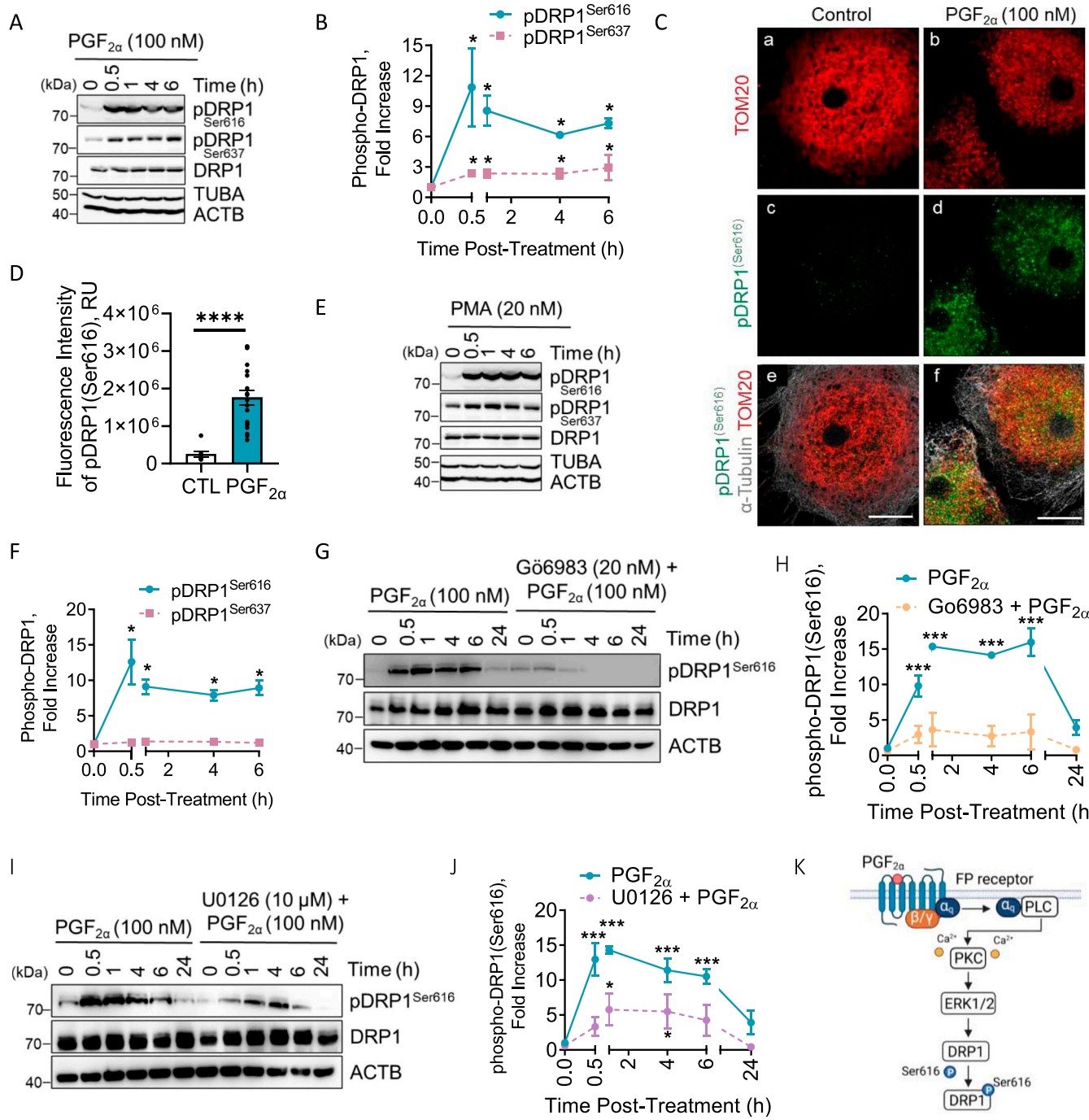

**Figure 3. Temporal effects of Prostaglandin F2alpha (PGF2α) on phosphorylation of dynamin-related protein-1 (DRP1) in vitro.**
Enriched bovine large luteal cells were treated with PGF2α (100 nM) or the PKC activator, Phorbol 12-myristate 13-acetate (PMA; 20 nM) for up to 24 h. Protein was extracted and subjected to Western blotting. **(A)** Representative Western blot of phosphorylation of DRP1 in large luteal cells after incubation with PGF2α. **(B)** Densitometric analyses of phospho-DRP1 after the stimulation with PGF2α. Statistics were performed by two-way ANOVA, which was used to evaluate repeated measures with Dunnett's post tests to compare means. Symbols represent mean fold changes (means ± sem, n = 3). Solid line: phospho-DRP1 (Ser616); dash line: phospho-DRP1 (Ser637). Enriched bovine large luteal cells were treated with PGF2α (100 nM) for 30 min and subjected to confocal microscopy. **(C)** Representative micrographs illustrating the effects of PGF2α on phosphorylation of DRP1 (Ser616) in large luteal cells. **(D)** Quantitative analyses of the mean fluorescence intensity (relative units; RU) of phospho-DRP1 (Ser616). Statistics were performed by t tests to evaluate paired responses. Open bars represent control cells; Closed bars represent 30 min posttreatment with PGF2α. **(E)** Representative Western blot of phosphorylation of DRP1 in large luteal cells after incubation with PMA. **(F)** Densitometric analyses of phospho-DRP1 after stimulation with PMA. Statistics were performed by two-way ANOVA was used to evaluate repeated measures with Dunnett's post tests to compare means. Symbols represent mean fold changes (means ± sem, n = 3). Solid line: phospho-DRP1 (Ser616); dash line: phospho-DRP1 (Ser637). Large bovine luteal cells were pretreated with PKC inhibitor, Go6983 (20 nM) or MEK1 inhibitor, U0126 (10 µM) for one h and subsequently treated with PGF2α (100 nM) for 0, 0.5, 1, 4, 6, or 24 h. Protein was extracted and subjected to Western blotting. **(G)** Representative Western blot analysis of phospho-DRP1 (Ser616) in large luteal cells pretreated with Go6983

During the early stages of luteolysis, PGF2α stimulates an increase in inflammatory cytokines, such as TNFα, interleukins (IL-1β, IL-6, IL-17A, and IL-33), and cytokine signaling intermediates (NF-κB, STAT), all of which may contribute to luteal regression (45). Large luteal cells were treated with TNFα for up to 24 h to determine the effects of increased luteolytic cytokine signaling on the phosphorylation of DRP1 at Ser616. Stimulating large luteal cells with TNFα increased phosphorylation of DRP1 (Ser616; 2.9-fold) 30 min posttreatment and remained elevated throughout the experimental period when compared with control (P < 0.05; Fig S3A and B).

As alluded to in previous section, the luteolytic actions of PGF2α manifest through receptor-mediated stimulation via the PKC signaling pathway and activation of downstream protein kinases. Large luteal cells were treated with PMA (20 nM; Fig 3E) for up to 24 h to determine the temporal nature of the phosphorylation of DRP1. Treatment with PMA, a PKC activator, increased phosphorylation of DPR1 (Ser616; 12-fold) 30 min posttreatment and remained elevated throughout the experimental period compared with control (P < 0.05; Fig 3E and F). In contrast to PGF2α, PMA did not influence the phosphorylation of DRP1 at Ser637 when compared with control (P > 0.05; Fig 3E and F). Similarly, PMA increased the mean fluorescent intensity of phospho-DRP1 (Ser616; 15-fold) when compared with control cells (P < 0.05; Fig S4A and B).

To determine the role of PKC/MAPK signaling on the phosphorylation of DRP1 at Ser616, enriched populations of large luteal cells were treated up to 24 h with PGF2α (100 nM) in the presence or absence of commercially available small molecule inhibitors (Fig 3G and I). PKC inhibitor (Go6983; 20 nM) abrogated the stimulatory effects of PGF2α treatment on the phosphorylation of DRP1 at Ser616 (P < 0.05; Fig 3G and H). ERK1/2 inhibition via MEK1/2 inhibitor (U0126; 10 μM) inhibited PGF2α-induced phosphorylation of DRP1 on Ser616 (P < 0.05; Fig 3I and J). In addition to activating PKC/ERK1/2 signaling cascade, PGF2α also promotes the phosphorylation of JNK and p38 MAPK (46). Drug inhibition of both JNK (SP600125; 20 μM) and p38 MAPK (SB207580; 10 μM) had no effect on PGF2α-induced phosphorylation of DRP1 at Ser616 (data not shown). Fig 3K illustrates the proposed model for PGF2α-induced phosphorylation of DRP1 on Ser616 in bovine large luteal cells.

### Effects of PGF2α on the phosphorylation of MFF

MFF is an outer mitochondrial membrane protein that binds phosphorylated DRP1 (Ser616) to promote fission of mitochondria. Large luteal cells were treated with PGF2α (100 nM; Fig 4A) or TNFα (10 ng/ml; Fig S3A) for up to 24 h to determine the temporal nature of the phosphorylation of MFF at Ser146, in vitro. Western blot revealed that PGF2α promotes the phosphorylation of MFF (Ser146; 1.7-fold; P < 0.01) 1 h posttreatment and remains elevated for at least

6 h when compared with control (P < 0.001, Fig 4A and B). Immunostaining for phosphorylated MFF after acute stimulation with PGF2α in bovine luteal cells is shown in Fig 3C. PGF2α rapidly increased the fluorescent intensity of phospho-MFF (Ser146; sevenfold) when compared with control cells (P < 0.0001; Fig 4C and D). Together, these results provide strong evidence to support our hypothesis that PGF2α acutely influences mitochondrial dynamics. Large luteal cells were also treated with TNFα for up to 24 h to determine the effects of increased luteolytic cytokine signaling on the phosphorylation of MFF at Ser146. Stimulating large luteal cells with TNFα increased phosphorylation of MFF (Ser146; 1.4-fold) 30 min posttreatment and remained elevated for 6 h when compared with control (P < 0.05; Fig S3A and C).

To determine whether phosphorylated DRP1 (Ser616) translocates to the mitochondria after PGF2α treatment, mixed luteal cell cultures were stimulated with PGF2α (100 nM) or PMA (20 nM) for 1 h and mitochondria were immediately isolated. Cultures of dispersed mixed luteal cells were used for mitochondrial isolation because of low cell yield after enrichment of large luteal cells. After incubation, the cells were fractionated, and aliquots of total cell lysate, cytosolic, and mitochondrial fractions were subject to Western blotting. We observed abundant nuclear factor of kappa light polypeptide gene enhancer in B-cell inhibitors, alpha (IκBα) and alpha-tubulin (TUBA) in the cytosolic fractions but not mitochondrial fractions (Fig 4E). We also observed abundant mitochondrial proteins (MFF, STAR, CYP11A1, VDAC, and TOM20) in the mitochondrial fraction but not cytosolic fractions (Fig 4E). Immunodetectable phospho-DRP1 (Ser616) was observed in the fraction containing isolated mitochondria after treatment with PGF2α and PMA (Fig 4E) after increasing exposure time.

The mitochondrial receptor MFF was recently was identified as a downstream substrate of AMPK signaling (32). To determine the role of AMPK signaling on the phosphorylation of MFF at Ser146, enriched populations of large luteal cells were pretreated for 60 min with compound C (50 μM), an inhibitor of AMPK signaling, and subsequently treated with PGF2α (100 nM) or AMPK activator, 5-aminoimidazole-4-carboxamide-1-β-D-ribofuranoside (AICAR; 1 mM) for 4 h (Fig 4F). PGF2α and AICAR increased the phosphorylation of AMPK (Thr172) 1.4- and 1.8-fold, respectively, 4 h posttreatment when compared with control (P < 0.05; Fig 4F and G). In addition, PGF2α and AICAR both increased phosphorylation of MFF (Ser146) 1.4-and 1.8-fold 4 h posttreatment when compared with control (P < 0.05; Fig 4F and H). AMPK inhibitor (Compound C) abrogated the stimulatory effects of PGF2α and AICAR treatment on the phosphorylation of MFF at Ser146 (P < 0.05; Fig 4F and H). Together, this indicates that PGF2α -induced activation of MFF is dependent on AMPK signaling (Fig 4I).

---

and stimulated with PGF2α. **(H)** Densitometric analyses of phospho-DRP1 (Ser616). Statistics were performed by two-way ANOVA, which was used to evaluate repeated measures with Dunnett's post tests to compare means. Symbols represent mean fold changes (means ± sem, n = 3). Solid line: PGF2α; dash line: Go6983 and PGF2α. **(I)** Representative Western blot analysis of phospho-DRP1 (Ser616) in large luteal cells pretreated with U0126 and stimulated with PGF2α. **(J)** Densitometric analyses of phospho-DRP1 (Ser616). Statistics were performed by two-way ANOVA, which was used to evaluate repeated measures with Dunnett's post tests to compare means. Symbols represent mean fold changes (means ± sem, n = 2). Solid line: PGF2α; Dash line: U0126 and PGF2α. **(K)** Illustration of PGF2α/PKC/ERK1/2 -induced phosphorylation of DRP1 (Ser616). Micron bar represents 20 μm. Significant difference between treatments compared with control, *P < 0.05; ***P < 0.001; ****P < 0.0001. Source data are available for this figure.

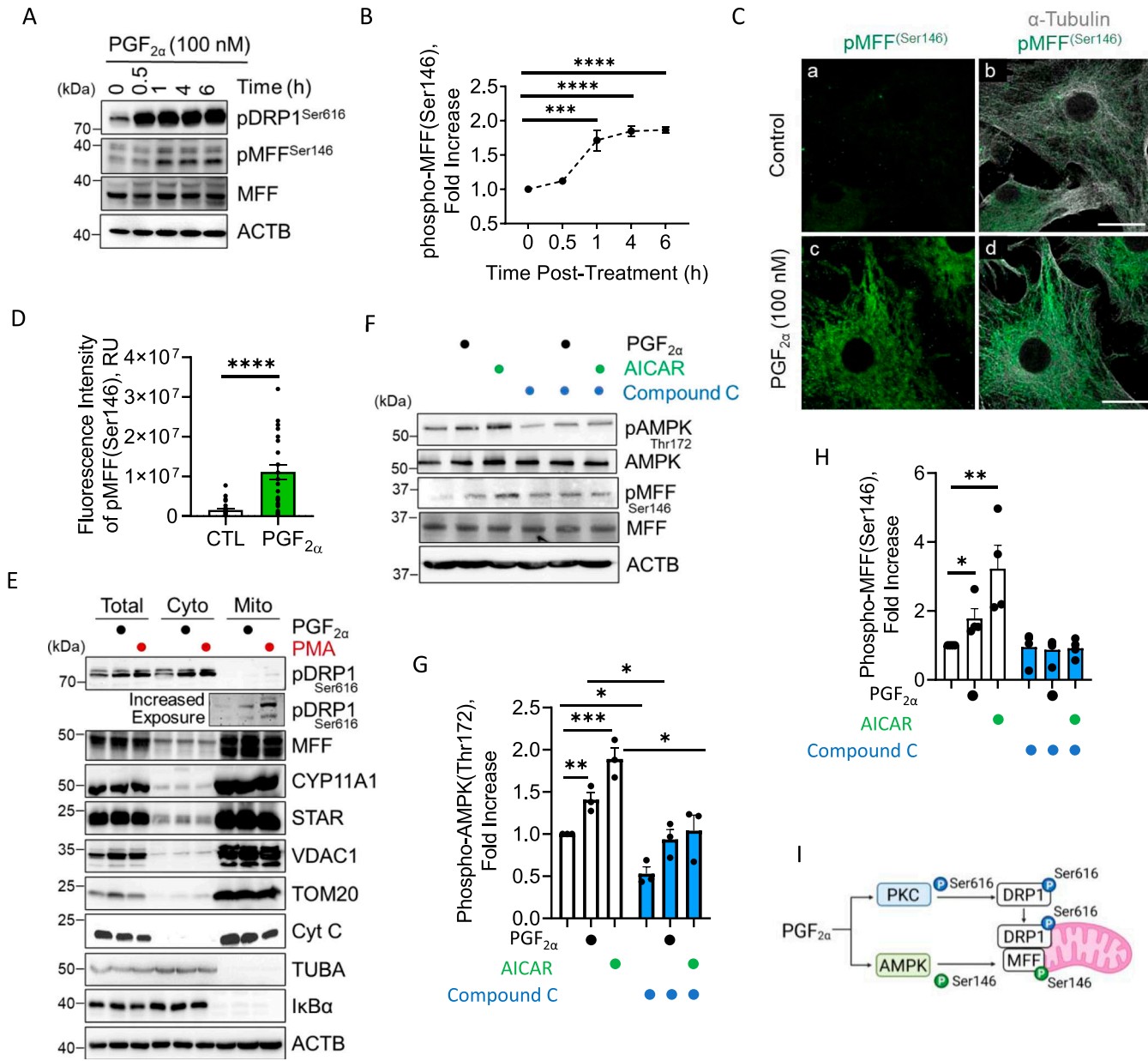

**Figure 4. Temporal effects of Prostaglandin F2alpha (PGF2α) on phosphorylation of mitochondrial fission factor (MFF) and localization of dynamin-related protein-1 (DRP1) in vitro.**
Large bovine luteal cells were treated with PGF2α (100 nM) for up to 24 h. Protein was extracted and subjected to Western blotting. **(A)** Representative Western blot analysis of phosphorylation MFF (Ser146) in large luteal cells treated with PGF2α. **(B)** Densitometric analyses of phospho-MFF (Ser146). Statistics were performed by one-way ANOVA followed by Dunnett's post tests to compare means. Symbols represent mean fold changes (means ± sem, n = 3). Large luteal cells were treated with PGF2α (100 nM) for 30 min and subject to confocal microscopy. **(C)** Representative micrographs of phosphorylation of MFF (Ser146) in large luteal cells after treatment with PGF2α. **(D)** Quantitative analyses of the fluorescence intensity (relative units; RU) of phospho-MFF (Ser146) after 30 min of incubation with PGF2α. Statistics were performed by t tests to evaluate paired responses. Mixed luteal cells were treated with PGF2α (100 nM) or PMA (20 nM), for 60 min. After treatment, mixed luteal cells were fractionated and resolved on Western blotting. **(E)** Representative Western blot of phospho-DRP1 (Ser616) from isolated mitochondria. Large luteal cells were pretreated for 60 min with compound C (50 μM) and subsequently treated with PGF2α (100 nM) or AMPK activator, 5-aminoimidazole-4-carboxamide-1-β-D-ribofuranoside (AICAR; 1 mM) for 4 h. Protein was extracted and subject to Western blotting. **(F)** Representative Western blot of phospho-AMPK (Thr172) and phospho-MFF (Ser146) obtained from large luteal cells treated with PGF2α or AICAR in the presence or absence of compound C. **(G)** Densitometric analysis of phospho-AMPK (Thr172). **(H)** Densitometric analysis of phospho-MFF (Ser146). Statistics were performed by two-way ANOVA, which was used to evaluate repeated measures with Dunnett's post tests to compare means. **(I)** Illustration of PGF2α-induced phosphorylation of DRP1 and MFF. The micron bar represents 20 μm. Significant difference between treatments compared with control, **$P < 0.05$; **$P < 0.01$; ***$P < 0.001$; ****$P < 0.0001$.
Source data are available for this figure.

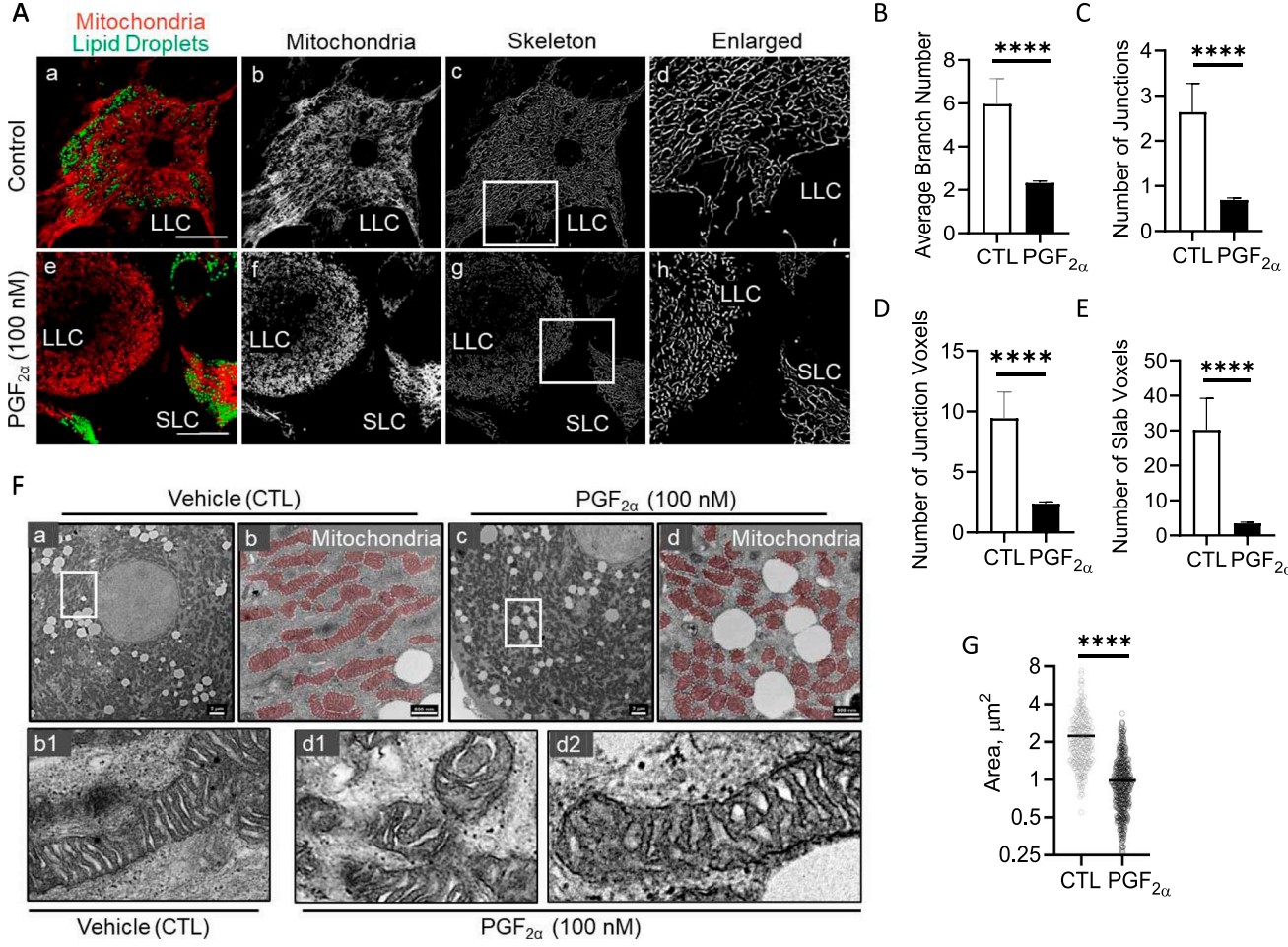

**Figure 5. Prostaglandin F2alpha (PGF2α) stimulates mitochondrial fission in large luteal cells.**
Luteal cells were treated with PGF2α (100 nM) for 1 h. Succeeding treatment, confocal microscopy was used to determine the influence of PGF2α on mitochondrial morphology. Representative micrographs (from left to right) of mitochondria (MitoTracker Red FM) and lipid droplets (BODIPY493/503; panels (a, e)), mitochondria (white; panels (b, f)), skeleton of mitochondria (panel (c, g)), and enlarged skeleton of mitochondria (panels (d, h)) obtained from cells treated (from top to bottom) with vehicle control or PGF2α (100 nM). **(B, C, D, E)** Quantitative analysis of branch number, (C) number of junctions, (D) number of junction voxels, and (E) number of slab voxels in cells treated with PGF2α for 1 h. Bars represent means ± sem; n = 3. **(F)** Enriched large luteal cells were treated with PGF2α (100 nM) for 4 h. Succeeding treatment, transmission electron microscopy (TEM) was used to determine the influence of PGF2α on mitochondrial morphology and size. Representative TEM micrographs were obtained from enriched large luteal cells treated (from top to bottom) with either control (panels (a, b, b1)) or PGF2α (panels (c, d, d1, d2)). Images were enlarged to examine the cristae morphology of individual mitochondria. **(G)** Quantitative analysis of the mitochondrial area in cells treated with PGF2α for 4 h. Representative confocal image, micron bar = 20 $\mu$m. Representative TEM image is shown at 21,000x and 35,900x, respectively, scale bar = 2 $\mu$m and 500 nm. Statistics were performed by $t$ tests to evaluate paired responses. Significant difference between treatments compared with control, $*P < 0.05$; $**P < 0.01$; $***P < 0.001$; $****P < 0.0001$.

### Effects of PGF2α on mitochondrial morphology

Because our data strongly support the hypothesis that PGF2α acutely influences mitochondrial dynamics through phosphorylation of DRP1 and MFF, we set out to determine the effects of PGF2α on mitochondrial morphology using confocal microscopy (Fig 5A). Mitochondrial branches vary from separated structures to interconnected networks. To determine the effect of PGF2α on both mitochondrial structure and network, we measured the branch number and number of junctions, junction voxels (if they have more than two neighbors), and slab voxels (if they have exactly two neighbors) (Fig S5). Large luteal cells treated with PGF2α had decreased number of branches ($P < 0.0001$; Fig 5B), junctions ($P < 0.0001$; Fig 5C), junction voxels ($P < 0.001$; Fig 5D), and slab voxel ($P < 0.0001$; Fig 5E), indicative of smaller individual-like mitochondria.

Transmission electron microscopy was employed to observe PGF2α-induced morphological changes to mitochondrial size and cristae organization (Fig 5F). Under basal conditions, large luteal cells have large, elongated mitochondria (Fig 5F panels a and b). Moreover, these mitochondria appear to have tightly packed, organized cristae junctions (Fig 5F panel b1). Inversely, large luteal cells treated with PGF2α had decreased mitochondrial area ($P < 0.0001$; Fig 5F panels c and d and 5G) compared with control large luteal cells. Moreover, cristae organization of large luteal cells treated with PGF2α appeared disrupted. PGF2α-treated large luteal cells contained more uncoupled cristae junctions, which were observed in both smaller individual-like mitochondria (Fig 5F panel d1) and elongated mitochondria (Fig 5F panel d2).

Mdivi-1 is a highly efficient small molecule inhibitor of mitochondrial fission. Mdivi-1 binds to an allosteric site blocking conformational

change necessary for DRP1 self-assembly and GTP hydrolysis required for mitochondrial division (47). To investigate whether DRP1 self-assembly with the mitochondria is required for PGF2α-induced mitochondrial fission, large luteal cells were pretreated with Mdivi-1 (5 µM) for 1 h and stimulated with PGF2α (100 nM) for up to 24 h (Fig S6A). Mdivi-1 had no influence on PGF2α-induced phosphorylation of DRP1 at Ser616 (P > 0.05; Fig S6A and B). Despite the phosphorylation state of DRP1, drug inhibition of DRP1 self-assembly with the mitochondria using Mdivi-1 attenuates PGF2α-induced mitochondrial fission (P < 0.0001; Fig S6C and D), supporting the hypothesis that PGF2α promotes activation and translocation of DRP1 in luteal cells, facilitating the regulation of mitochondrial dynamics.

### Effects of PGF2α on the production of ROS in large luteal cells

ROS are highly reactive molecules that, if unchecked, can cause intracellular damage (48). Previous studies indicate that activation of PKC via PGF2α stimulates the accumulation of ROS that facilitate luteolysis (49); however, this process is not fully understood. To determine the effects of PGF2α on ROS production, mixed luteal cells were stimulated with PGF2α (100 nM) for 4 h and ROS was visualized by confocal microscopy using CellROX Green Reagent (Fig 6A). Upon oxidation by ROS, CellROX subsequently binds to DNA, promoting aggregated bright green photostable fluorescence for optimal detection. ROS present in cytoplasmic compartments accumulate on nuclear DNA, whereas mitochondrial ROS convene on mitochondrial DNA (Fig 6B). PGF2α substantially increased ROS production compared with control cells (P < 0.0001; Fig 6A and C). Interestingly, in large luteal cells, ROS production remained confined within mitochondrial compartments as shown by punctate staining localized at mitochondrial DNA. Moreover, PGF2α stimulated ROS production in small and non-steroidogenic luteal cells (Fig 6A) despite low abundance of PGF2α receptors (42, 43), suggesting a potential paracrine signaling mechanism initiated by the large luteal cells.

### Effects of PGF2α and AMPK on activation of mitophagy in large luteal cells

Mitochondrial fission and mitophagy are two cellular mechanisms that synchronously regulate mitochondrial quality control systems to protect cells from cytotoxic ROS production (50). AMPK is an enzyme that plays a role in energy homeostasis and is a downstream target of PGF2α in bovine luteal cells (51). AMPK is also a regulator of autophagy and mitophagy through activation of the protein kinase, Unc-51-like autophagy activating kinase (ULK1) (32). To investigate the effects of PGF2α and AMPK on the phosphorylation of proteins involved in mitochondrial fission and mitophagy, large luteal cells were treated with PGF2α (100 nM) or the AMPK activator, AICAR (1 mM), for 4 h, and subject to Western blotting (Fig 6D). PGF2α and AICAR increased the phosphorylation of AMPK (Thr172) 1.5 and 2.2-fold, respectively, 4 h posttreatment when compared with control (P < 0.05; Fig 6D and E). AICAR had no effect on the phosphorylation of DRP1 (Ser616) 4 h posttreatment when compared with control (P > 0.05; Fig 6D and F). In addition, PGF2α and AICAR both increased phosphorylation of MFF (Ser146) 1.4-fold

4 h posttreatment when compared with control (P < 0.05; Fig 6D and G).

We and others have reported that MFF is a downstream substrate of AMPK signaling (32). Moreover, unphosphorylatable MFF mutants have been reported to block mitophagy (52), putatively connecting PGF2α/AMPK to mitochondrial fission and to mitophagy. Here, we identified mitophagy-associated proteins that were phosphorylated in large luteal cells in response to treatment with PGF2α or AICAR. We observed a 1.4- and 1.7-fold increase in the phosphorylation of PINK1 at Ser228 posttreatment with PGF2α and AICAR, respectively, when compared with control (P < 0.05; Fig 6D and H). We also observed a 1.2- and 1.4-fold increase in the phosphorylation of Parkin at Ser65 posttreatment with PGF2α and AICAR, respectively, when compared with control (P < 0.05; Fig 6D and I). In addition, PGF2α and AICAR both increased phosphorylation of ULK1 (Ser555) 1.6-fold 4 h posttreatment when compared with control (P < 0.05; Fig 6D and J). Lastly, PGF2α and AICAR increased the levels of LC3B protein by 1.6- and 1.4-fold, respectively, 4 h posttreatment when compared with control (P < 0.05; Fig 6D and K).

### Temporal effects of PGF2α on activation of mitophagy in large luteal cells

To determine the effects of PGF2α on the phosphorylation of proteins involved in mitophagy, large luteal cells were treated with PGF2α (100 nM) for up to 4 h and subject to Western blotting (Fig 7A). PGF2α induced an acute 1.8-fold increase in the phosphorylation of AMPK (Thr172) within 30 min of treatment when compared with control (P < 0.01) and remained elevated 4 h posttreatment in large luteal cells (P < 0.01; Fig 7A and B). PGF2α increased the phosphorylation of PINK1 (Ser228) by 1.4-fold (P < 0.001) within 30 min of treatment and 1.8-fold for 1 and 4 h posttreatment when compared with control (P < 0.05; Fig 7A and C). Lastly, PGF2α increased phosphorylation of ULK1 (Ser555; 1.6-fold) 4 h posttreatment when compared with control (P < 0.01; Fig 7A and D).

To determine the effects of PGF2α on mitophagy activation, we stimulated enriched populations of large luteal cells with PGF2α (100 nM) for 6 h and visualized mitophagy by confocal microscopy using Mtphagy Dye (Fig 7E and F). Under basal conditions, Mtphagy Dye accumulates and is immobilized in intact mitochondria and exhibits a weak fluorescence. When Mtphagy is induced, the damaged mitochondria fuse to lysosomes and the dye emits a high fluorescence (Fig 7E). Treatment with PGF2α for 6 h increased the fluorescent intensity of the Mtphagy Dye by 3.4-fold when compared with control cells (P < 0.01; Fig 7F and G). Lysosomes were co-stained using Lyso dye to confirm the fusion of Mtphagy Dye-labeled mitochondria and lysosomes. Treatment with PGF2α for 6 h increased the colocalization of lysosomes with the Mitophagy dye 2.4-fold when compared with control cells (P < 0.0001; Fig 7F and H).

## Discussion

In the present study, we delineate the effects of the luteolytic hormone, PGF2α, on mitochondrial dynamics and activation of mitophagy in bovine luteal cells. To our knowledge, the present

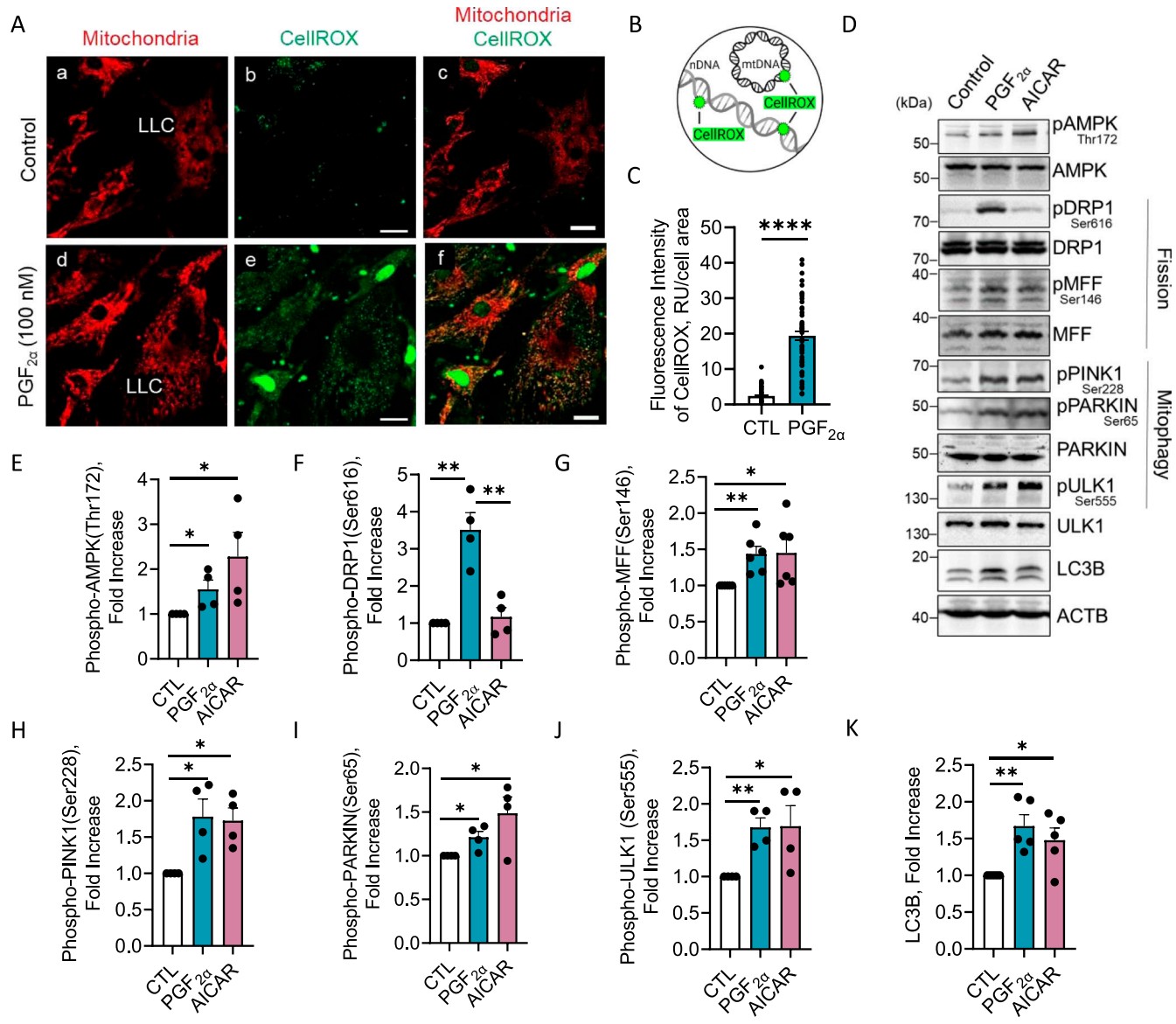

**Figure 6. Effects of Prostaglandin F2alpha (PGF2α) on reactive oxygen species (ROS) production and activation of mitophagy in bovine large luteal cells in vitro.**
Mixed luteal cells were stimulated with PGF2α (100 nM) for 4 h, and confocal analysis was used to visualize ROS production. **(A)** Representative micrographs showing the effects of PGF2α on ROS production and localization in luteal cells. From left to right; mitochondria (MitoTracker Red FM; white; panels (a, d)), CellROX (panels (b, e)), and merger of mitochondria and CellROX (panel (c, f)) obtained from cells treated with control or PGF2α (100 nM). Nuclear fluorescence represents cytoplasmic ROS. Mitochondrial fluorescence represents mitochondrial ROS. Large luteal cell. **(B)** Illustration of CellROX Green Reagent mechanism. CellROX is a weakly fluorescent cell-permeant dye that exhibits bright green photostable fluorescence upon oxidation by ROS and subsequently binds to DNA. Nuclear DNA. Mitochondrial DNA. **(C)** Quantitative analyses of the fluorescence intensity normalized to cell area (relative units; RU) of CellROX. Statistics were performed by $t$ tests to evaluate paired responses. The open bar represents control cells; the closed bar represents cells treated with PGF2α. Enriched populations of large luteal cells were stimulated with PGF2α (100 nM) or AMPK activator, 5-aminoimidazole-4-carboxamide-1-$\beta$-D-ribofuranoside (AICAR; 1 mM) for 4 h. Protein was extracted and subjected to Western blotting. **(D)** Representative Western blot analysis of the phosphorylation of proteins associated with mitochondrial fission and mitophagy in large luteal cells four h posttreatment with PGF2α or AICAR. **(E)** Densitometric analysis of phospho-AMPK (Thr172). **(F)** Densitometric analysis of phospho-DRP1 (Ser616). **(G)** Densitometric analysis of phospho-MFF (Ser146). **(H)** Densitometric analyses of phospho-PINK1 (Ser228). **(I)** Densitometric analyses of phospho-Parkin (Ser65). **(J)** Densitometric analyses of phospho-ULK1 (Ser555). **(K)** Densitometric analyses of LC3B. Symbols represent mean fold changes (means ± sem, n = 3–4). The open bar represents control cells; the black closed bar represents cells treated with PGF2α; the grey closed bar represents cells treated with AICAR. Statistics were performed by one-way ANOVA followed by Tukey's multiple comparison tests. Significant difference between treatments compared with control, $*P < 0.05$; $**P < 0.01$; $****P < 0.0001$. Micron bar represents 20 $\mu$m.

study provides the first demonstration in any tissue that PGF2α signaling (1) stimulates the phosphorylation of DRP1 and MFF, (2) stimulates mitochondrial fission, (3) promotes an increase in intracellular ROS production, and (4) activates mitophagy of damaged mitochondria. These findings indicate that DRP1 is a PGF2α/PKC-sensitive molecule, and both mitochondrial dynamics and mitophagy are targets of PKC and AMPK signaling in large luteal cells during the initiation of luteolysis.

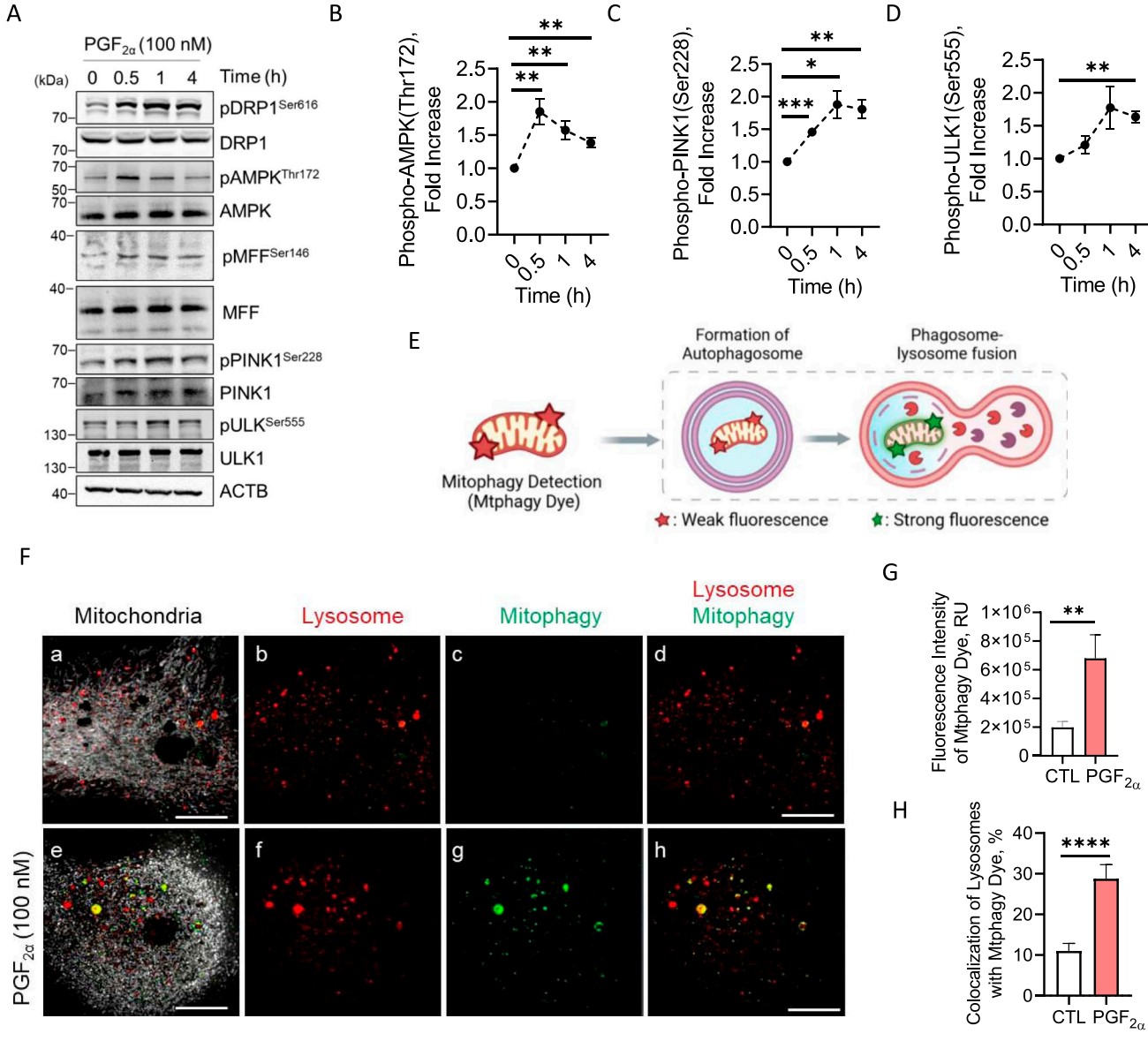

**Figure 7. Temporal effects of Prostaglandin F2alpha (PGF2α) on activation of mitophagy machinery in vitro.**
Enriched populations of large luteal cells were stimulated with PGF2α (100 nM) for 4 h and subject to Western blotting to detect activation of mitophagy. **(A)** Representative Western blot analysis of the phosphorylation of DRP1, AMPK, MFF, PINK1, ULK1, and proteins involved in activation of mitophagy in large luteal cells 4 h posttreatment with PGF2α. **(B)** Densitometric analyses of phospho-AMPK (Thr172). **(C)** Densitometric analyses of phospho-PINK1 (Ser228). **(D)** Densitometric analyses of phospho-ULK1 (Ser555). Statistics were performed by one-way ANOVA followed by Dunnett's post tests to compare means. Symbols represent mean fold changes (means ± sem, n = 3). Enriched populations of large luteal cells were stimulated with PGF2α (100 nM) for 6 h and confocal analysis was used to visualized mitophagy. **(E)** Illustration of the Mtphagy Dye (Mtphagy) mechanism. Mitophagy dye is a weakly fluorescent dye that segregates and immobilizes to the mitochondria and exhibits bright photostable fluorescence upon fusion with lysosomes. **(F)** Representative micrographs showing the effects of PGF2α on activation of mitophagy. From left to right; mitochondria (MitoTracker Red FM; white; panels (a, e)), lysosomes (Lyso Dye; panels (b, f)), mitophagy (Mtphagy; panel (c, g)), and co-localization of Mtphagy Dye reagent with lysosomes (panels (d, h)) obtained from cells treated with control or PGF2α (100 nM). **(G)** Quantitative analyses of the fluorescence intensity of the Mtphagy Dye. **(H)** Quantitative analysis of the localization of lysosomes with the Mtphagy. Statistics were performed by t tests to evaluate paired responses. The open bar represents control cells; the closed bar represents cells treated with PGF2α. Micron bar represents 20 μm. Significant difference between treatments compared with control, *$P < 0.05$; **$P < 0.005$; ***$P < 0.001$; ****$P < 0.0001$.

The fate of the corpus luteum is governed by luteotrophic (i.e., LH via cAMP/PKA signaling) or luteolytic hormones (i.e., PGF2α via PKC/MAPK and AMPK signaling). We recently reported that LH, via the PKA signaling pathway, reduces DRP1 activity and association with mitochondria, thus stabilizing mitochondria to steroidogenesis, a process required for progesterone synthesis in luteal cells (53).

Here, we coupled in vivo and in vitro approaches using bovine corpora lutea, to better understand the role of luteolytic PGF2α on the phosphorylation and activation of mitochondrial fission proteins, DRP1 and MFF. We observed that administration of PGF2α in vivo rapidly stimulates the phosphorylation of both DRP1 at Ser616 and MFF at Ser146. Moreover, immunostaining of

phosphorylated DRP1 and MFF appeared predominantly within the large luteal cells after administration of PGF2α. Using enriched populations of large luteal cells, we report that PGF2α-induced phosphorylation of DRP1 at Ser616 is dependent on activation of PKC/ERK signaling, whereas phosphorylation of its cognate mitochondrial receptor, MFF, is dependent on PGF2α-induced activation of AMPK. This is consistent with others who have reported PKC/ERK-dependent phosphorylation of DRP1 (29, 54, 55) and enhanced phosphorylation of MFF by AMPK (32). Together, these findings highlight that integration of PKC/ERK and AMPK signaling pathways are necessary for PGF2α-induced mitochondrial fission in large luteal cells.

Mitochondria undergo continual cycles of fusion and fission to regulate morphology and meet energy demands. Steroidogenesis is a complex process that requires the fusion of mitochondria to precede (56). Mitochondria can modulate their functions by switching from elongated interconnected networks to a fragmented state, allowing for complex quality control. Mitochondria also reorganize their internal structure by modifying cristae shape and organization (57). We provide evidence that PGF2α provokes a shift in mitochondrial dynamics via phosphorylation and activation of DRP1 and MFF. Using high-resolution confocal microscopy and electron microscopy, we observed that PGF2α promotes the translocation of cytoplasmic DRP1 to the mitochondria and stimulates fission of mitochondria. Inhibiting interactions between DRP1 and MFF using the commercially available drug inhibitor, Mdivi-1, attenuated PGF2α-induced mitochondrial fission without influencing the phosphorylation status of DRP1. Here, we report PGF2α also induced morphological changes to the internal mitochondrial structures, resulting in widening of cristae, uncoupling from cristae junctions, and formed condensed dilated compartments. Cristae remodeling is a dynamic process that can be induced in response to altered physiological or metabolic ques (58). The ability of DRP1 to rapidly associate with and act on the mitochondria after PGF2α treatment opens a new understanding toward the role of mitochondrial dynamics in the corpus luteum.

Mitochondria represent a major source of intracellular ROS generation. ROS are highly reactive molecules that, if unchecked, can cause intracellular damage (48). PKC activation, via PGF2α, stimulates the accumulation of ROS and production of cytokines and chemokines (45), facilitating luteal arrest. Here, we report that PGF2α substantially increases ROS production in small and non-steroidogenic bovine luteal cells. This agrees with other studies that indicate that activation of PKC via PGF2α stimulates the accumulation of ROS that facilitate luteolysis (49). How ROS contributes to mitochondrial dynamics in luteal cells requires further investigation.

Mitochondrial fission and mitophagy are two cellular mechanisms that synchronously regulate mitochondrial quality control pathways and protect cells from cytotoxic ROS production (50). PINK1 acts upstream of Parkin in a concerted action to initiate the removal of damaged mitochondria (59). Autophosphorylation of PINK1 at Ser228 and Ser402 promote recruitment of cytosolic Parkin to the outer membrane of damaged mitochondria (60). We provide evidence that luteolytic PGF2α provokes a rapid accumulation of total and phosphorylated PINK1 at Ser228, both in vivo and in vitro. Although it was not directly measured, PGF2α likely leads to rapid

depolarization of mitochondria membranes, whereby inhibiting PINK1 degradation. Positive feedback between PGF2α-induced increases in cytosolic Ca2+ levels released from the ER and elevated mitochondrial ROS production could trigger the opening of mitochondrial permeability transition pores and promote depolarization of mitochondrial membranes. In addition to promoting activation of PINK1, PGF2α rapidly stimulated phosphorylation of Parkin at Ser65 in large luteal cells. Phosphorylation of ULK1 by AMPK initiates recruitment of the ULK1-complex to ubiquitinated mitochondria, resulting in engulfment into LC3-positive autophagosomes (61). We further report, both PGF2α and AICAR, AMPK activators, stimulate the phosphorylation of ULK1 at Ser555, a phospho-site that recruits ULK-complexes to the mitochondria. This was accompanied by an increase in LC3B expression and visualized using Mtphagy reagent (62). Regulation of mitochondrial quality control pathways may be an initial cellular process associated with the early stages of luteolysis.

The mechanisms involved in luteolysis are highly complex, species-specific, and not well understood. Here, in vivo administration of PGF2α results in a precipitous decline (52%) in systemic progesterone concentration 2 h after treatment. This is consistent with the current notion that serum progesterone concentrations fall in parallel with luteal blood flow during functional regression of the gland. Interestingly, intraluteal tissue progesterone concentrations were also decreased (54%) 4 h after PGF2α treatment, yet, no changes in steroidogenic enzymes, STAR, CYP11A1 or HSD3B, protein expression (Fig 1) or mRNA expression (45) were observed. Rapid changes in mitochondrial dynamics and energetics demands (53, 56), together with reduced ability to use cholesterol (45, 63), may influence luteal steroidogenic capacity, rapidly decreasing progesterone output without influencing the expression of key steroid synthesizing enzymes. We report for the first time that mitochondrial dynamics and initiation of mitophagy are a downstream target of PGF2α in bovine luteal cells. Furthermore, integration of PKC and AMPK signaling pathways are necessary for modulating PGF2α-induced activation of mitochondrial fission in large luteal cells. Though the precise role remains unknown, activation of mitochondrial quality control systems, that is, mitochondrial fission and mitophagy, are a key feature of early functional luteal regression and highlights new understanding toward the need for proper integration of PKC and AMPK signaling pathways in response to PGF2α.

A limitation of the current study is that this study examined the luteal response to a single luteolytic dose of a potent PGF2α analogue commonly used to regulate the reproductive cycle. However, physiological luteolysis occurs in response to multiple sequential pulses of uterine-derived PGF2α (64). In addition, in vitro studies, although allowing a detailed understanding of the response to PGF2α in luteal cells expressing the PTGFR, do not fully represent the complex cellular interactions among various cell types occurring in vivo, and thus do not fully represent the changes occurring in the luteal tissue microenvironment in response to PGF2α.

Luteolysis is a natural event necessary to regulate the female estrous cycle. An adequate corpus luteum function is, however, essential for the establishment and maintenance of pregnancy. Defects in luteal function are associated with implantation failure and premature termination of pregnancy. In the present study, we

provide evidence that luteolytic hormones modulate mitochondrial dynamics by increasing the phosphorylation and activity of DRP1 and MFF. Furthermore, we bring forward the notion that PGF2α induces mitochondrial fission and uncoupling of cristae junctions, a process that requires proper integration of PKC and AMPK signaling pathways. We highlight PKC/ERK as a key upstream regulator of DRP1 phosphorylation and AMPK as a crucial regulator associated with phosphorylation of MFF. In conjunction with shifts in mitochondrial dynamics, PGF2α triggers intracellular rises in ROS production and activation of mitophagy. Together, these findings signify that PGF2α de-stabilizes luteal mitochondria as a proximal event during luteolysis. Taken together, our findings place the mitochondria as a novel target downstream of PKC and AMPK signaling in response to the luteolytic lipid mediator, PGF2α. Understanding the cellular processes involved with early luteal regression may serve as a target for improving fertility.

The luteal phase is often overlooked in fertility research and the present findings may be relevant to human reproduction. Although the initial signal for luteolysis in non-human primates and women is a loss of gonadotropic support (65), studies provide evidence for an increase in intraluteal PGF2α production in response to a loss of gonadotropin support (66, 67). Similar to the bovine model, injection of PGF2α causes luteolysis in women (68) and non-human primates (69). Furthermore, gene expression profiles of regressing primate corpora lutea (70) are similar to profiles observed in regressing bovine corpora lutea (45, 71, 72). Insight into the mechanisms of luteolysis, including the actions of PGF2α, can aid in the development of targets for optimizing the length of the luteal phase for fertility in both cows and women.

# Materials and Methods

### Reagents

Penicillin G-sodium, streptomycin sulfate, HEPES, BSA, deoxyribonuclease l, FBS, Tris–HCl, sodium chloride, EDTA, EGTA, sodium fluoride, $Na_4O_2O_7$, $Na_3VO_4$, Triton X-100, glycerol, dodecyl sodium sulfate, $\beta$-mercaptoethanol, bromophenol blue, Tween-20, paraformaldehyde, phorbol 12-myristate 13-acetate (PMA), SP600125 (JNK inhibitor), and SB207580 (p38 MAPK inhibitor) were purchased from Sigma-Aldrich. The phosphate buffer solution, DMEM (Calcium-free, 4.0 g/l glucose), penicillin–streptomycin solution, Trypan Blue, Halt Protease, and Phosphatase Inhibitor Cocktail were purchased from Invitrogen Corporation (Thermo Fisher Scientific). The opti-MEM, M199 culture medium, and gentamicin sulfate were purchased from Gibco (Thermo Fisher Scientific). Collagenase was purchased from Atlanta Biologicals (Flowery Branch). Prostaglandin F2α was purchased from Cayman Chemical and bovine LH was purchased from Tucker Endocrine Research Institute. Recombinant TNFα was purchased from R&D systems. Go6983, PKC inhibitor was purchased from Abcam. No. 1 glass coverslips, microscope slides, and chemiluminescent substrate (SuperSignal West Femto) were from Thermo Fisher Scientific. Fluoromount-G and clear nail polish were purchased from Electron Microscopy Sciences. FSK and compound C was purchased from EMD Millipore. BCA protein assay and 4–20% Mini-PROTEAN TGX

precast protein gels were purchased from Bio-Rad and the nonfat milk was from a local Kroger. Mitochondrial isolation kit was purchased from QIAGEN (Cat. No 37612; Qproteome). Mdivi-1 and AICAR were purchased from Tocris. An ELISA kit for progesterone was purchased from DRG International, Inc. ImmuChemTM Coated Tube Progesterone 125I RIA kit was purchased from ICN Pharmaceuticals, Inc. Table 1 lists all antibodies used in the study.

### Part I: In vivo analysis

#### Cattle

Post-pubertal, non-lactating multiparous female cattle (n = 6) of composite breeding (25% MARC III [1/4 Angus, 1/4 Hereford, 1/4 Pinzgauer, 1/4 Red Poll], and 75% Red Angus) beef cows from the beef physiology herd at the Eastern Nebraska Research and Extension Center (ENREC), were used in this study. Cows were synchronized using two intramuscular injections of PGF2α (25 mg; Lutalyse, Zoetis Inc.) 11 d apart. At mid-cycle (days 9–10), the cows were treated with an intra-muscular injection of saline (n = 3) or PGF2α (25 mg; n = 9). At each of four time-points postinjection (0, 1, 2, and 4 h), three cows per treatment were subjected to a bilateral ovariectomy through a right flank approach under local anesthesia as previously described (45, 73, 74). The corpus luteum was removed from each ovary, weighed, and < 5 $mm^3$ sections were snap-frozen in liquid $N_2$ for subsequent protein analysis or fixed in 10% formalin for immunohistochemistry. The University of Nebraska–Lincoln Institutional Animal Care and Use Committee approved all procedures and facilities used in this animal experiment and animal procedures were performed at the University of Nebraska—Lincoln, Animal Science Department.

#### Progesterone analysis

Progesterone was extracted from luteal tissue homogenate using a double-extraction procedure before ELISA assay as described (75). Briefly, in duplicate, 50–100 mg of tissue and 1 ml petroleum ether were mixed in a 10 × 13 mm glass test tube. Phase separation was accomplished by placing samples into a −80°C freezer for 5 min. The organic phase was decanted into a clean glass test tube, and an additional 1 ml petroleum ether was added to the aqueous phase and phases were separated as above. Organic phases were combined and evaporated using $N_2$ gas. The samples were reconstituted in 1 × ELISA buffer at a 1:200 dilution.

Plasma progesterone concentrations were determined using a RIA to detect progesterone concentrations as previously described (76). Progesterone concentrations were determined using the ImmuChemTM Coated tube Progesterone 125I RIA kit (intra-assay CV = 5.64%, inter-assay CV=7.43%). The sensitivity of the kit is 0.02 ng/ml. Progesterone concentrations from luteal tissue homogenate was determined using a commercially available ELISA kit per manufacturer's protocol (intra-assay CV = 2.3%; one assay). The analytical sensitivity of the kit is 0.045 ng/ml.

#### Western blotting analysis

Approximately, 100 mg of tissue was homogenized in RIPA buffer supplemented with 1× Halt Protease and Phosphatase Inhibitor Cocktail and sonicated at 40% power setting (Model CV188; VibraCell) as previously described (53). After sonification, tissue homogenates

were centrifuged at 13,000$g$ at 4°C for 15 min. Protein was collected and concentrations were determined using BCA protein assay. Samples were suspended in 6× Laemmli buffer and placed on a dry heat bath at 100°C for 6 min.

Proteins (30 $\mu$g/sample) were resolved using 10% SDS–PAGE or 4–20% Mini-PROTEAN TGX precast protein gel and then transferred to nitrocellulose membranes. The membranes were blocked with Tris-buffered saline + 0.1% Tween-20 (TBS-T) containing 5% nonfat milk solution at room temperature for 1 h. The membranes were incubated in a primary antibody (Table 1) for 24 h at 4°C for detection of total and phosphorylated proteins. The membranes were rinsed three times with TBS-T for 5 min. The membranes were then incubated with appropriate horseradish peroxidase-linked secondary antibody (Table 1) for 1 h at room temperature. Blots were then rinsed with TBS-T, and a chemiluminescent substrate was applied per manufacturer's instructions. Blots were visualized using a UVP Biospectrum 500 Multi-Spectral imaging system (UVP) and the percent abundance of immunoreactive protein was determined using densitometry analysis in VisionWorks (UVP).

Total proteins were normalized to ACTB before calculation of fold induction. The ratio of phosphorylated DRP1 to total DRP1 was determined for each treatment and time point. Fold increases because of the treatment (control versus PGF2$\alpha$) were then calculated.

### Immunohistochemistry

Portions of ovaries containing corpora lutea obtained from cows treated with an intramuscular injection of saline (n = 3) or PGF2$\alpha$ (n = 3; 4 h) were fixed in 10% formalin for 24 h and then changed into 70% ethanol until embedded in paraffin. Tissues were cut into 4 $\mu$m sections and mounted onto polylysine-coated slides. Slides were deparaffinized through three changes of xylene and through graded alcohols to water and microwaved in unmasking solution (Vector H-3300) for antigen retrieval. Endogenous peroxidase was quenched with 0.3% hydrogen peroxide in methanol for 30 min. Sections were incubated with anti-DRP1, anti-phospho-DRP1 (Ser616 or Ser637), anti-phospho-MFF (Ser146) or anti-MFF as indicated in Table 1, and subsequently, anti-rabbit ABC (Vector PK-4001) and stained using a DAB detection kit (Vector SK-4100). Slides were counterstained with Mayer's hematoxylin, dehydrated through graded alcohols, and mounted with Fluoromount-G. Nonimmune IgG from the host species was used as control (Table 1).

### Part II: In vitro analysis

### Microarray analysis

We mined bovine gene expression arrays from NCBI GEO repository (GSE83524) to analyze the expression of steroidogenic machinery in freshly isolated bovine granulosa (GC, n = 4), large luteal (LLC, n = 3), and small luteal (SLC, n = 3) cells from mature corpora lutea. Details of the isolation and analysis were previously published (42, 43).

### Tissue collection, cell preparation, and elutriation

Bovine ovaries were collected at a local slaughterhouse from mid-cycle non-pregnant cows. Uteri were checked for presence of a fetus or visible gross abnormalities. The ovaries were immersed in 70% ethanol and then transported to the laboratory at 4°C in PBS.

Using sterile technique, the corpus luteum was surgically dissected from the ovary and finely minced and dissociated using collagenase (103 U/ml) in basal medium (M199 supplemented with antibiotics [100 U/ml penicillin G-sodium, 100 $\mu$g/ml streptomycin sulfate, and 10 $\mu$g/ml gentamicin sulfate]) for 45 min in spinner flasks at 35°C. The supernatant was transferred to a sterile 15 ml culture tube, washed three times with sterile PBS, and resuspended in 10 ml of elutriation medium (calcium-free DMEM medium, 4.0 g/l glucose, antibiotics, 25 mM HEPES, 0.1% BSA, and 0.02 mg/ml deoxyribonuclease l; pH 7.2) on ice. Fresh dissociation medium was added to the remaining undigested tissue and incubated with agitation for an additional 45 min and processed as described above. Viability of cells was determined using Trypan Blue and cell concentration was estimated using a hemocytometer before cell elutriation.

Freshly dissociated cells were resuspended in 30 ml elutriation medium. Dispersed luteal cells were enriched for small and large luteal cells via centrifugal elutriation as previously described (77). Cells with a diameter of 15–25 $\mu$m were classified as small luteal cells (purity of > 90% enriched small luteal cells) and cells with diameter > 30 $\mu$m were classified as large luteal cells (purity of 55–90% enriched large luteal cells).

### Cell preparation and treatments

Enriched populations of small and large luteal cell cultures were plated in 12-well culture dishes at $5 \times 10^5$ cells/well and $2 \times 10^5$ cells/well, respectively. Cells were cultured in culture media (M199 supplemented with 5% FBS, 0.1% BSA, and antibiotics) at 37°C in an atmosphere of 95% humidified air and 5% $CO_2$ as described above.

### Cell treatments

Before treatments, cells were rinsed with PBS and fresh culture medium was placed on cells and equilibrated at 37°C in atmosphere of 95% air and 5% $CO_2$ for 2 h.

To determine the influence of luteotrophic and luteolytic hormones on the differential phosphorylation of DRP1, enriched populations of small and large luteal cells were treated with the culture medium alone, FSK (10 $\mu$M; FSK), PGF2$\alpha$ (100 nM), phorbol 12-myristate 13-acetate (20 nM; PMA), TNF$\alpha$ (10 ng/ml; TNF$\alpha$) or luteinizing hormone (10 ng/ml; LH) for 30 min at 37°C in atmosphere of 95% air and 5% $CO_2$.

For time response experiments with drug inhibitors (Go6983, U0126 or Mdivi-1), cells were pretreated for 1 h and then subsequently treated with culture medium alone or PGF2$\alpha$ (100 nM) for 0, 0.5, 1, 4, 6 or 24 h at 37°C in an atmosphere of 95% air and 5% $CO_2$

### Western blotting analysis

After incubation, cells were immediately placed on ice and rinsed three times with 1 ml of ice-cold PBS. The cells were lysed with 50 $\mu$l cell lysis buffer and removed from the culture dish using a cell scraper for sonication at 40% power setting (Model CV188; VibraCell) as previously described (53). Samples were suspended in 6× Laemmli buffer and placed on a dry heat bath at 100°C for 6 min.

Proteins (20 $\mu$g/sample) were resolved and visualized as described in Part 1. Total proteins were normalized to ACTB or tubulin before calculation of fold induction. The ratio of phosphorylated DRP1 to total DRP1 was determined for each treatment and time

**Table 1.   Characteristics of antibodies used for Western blotting and microscopy.**

| Antibody name | Dilution ratio | Species specificity | Source | Supplier | Cat. No |
|---|---|---|---|---|---|
| Phospho-DRP1 (Ser637) | 1:1,000[a]/1:200[b] | Mouse | Rabbit mAB | Cell Signaling | 4867S |
| Phospho-DRP1 (Ser616) | 1:1,000[a]/1:200[b] | Human | Rabbit mAB | Cell Signaling | 4494S |
| DRP1 | 1:1,000[a]/1:200[b]/1:200[c] | Mouse | Rabbit mAB | Cell Signaling | 8570S |
| Phospho-MFF (Ser146) | 1:1,000[a]/1:200[b]/1:200[c] | Mouse | Rabbit mAB | Cell Signaling | 49281 |
| MFF | 1:1,000[a]/1:200[b] | Mouse | Rabbit pAB | Cell Signaling | 86668S |
| HSL | 1:1,000 | Mouse | Rabbit pAB | Cell Signaling | 4107 |
| STAR | 1:10,000 | Mouse | Rabbit pAB | Abcam | ab96637 |
| CYP11A1 | 1:1,000 | Mouse | Rabbit mAB | Cell Signaling | 14217 |
| HSD3B | 1:1,000 | Mouse | Rabbit mAB | A gift from Dr. Ian Mason | |
| TOM20 | 1:200[c] | Mouse | Rabbit mAB | Cell Signaling | 42406S |
| VDAC1/Porin | 1:1,000 | Mouse | Rabbit pAB | Abcam | Ab15895 |
| IkBα | 1:1,000 | Mouse | Rabbit pAB | Santa Cruz Biotechnology, Inc. | Sc-847 |
| Cyt C | 1:1,000 | Bovine | Mouse mAB | Abcam | ab110325 |
| Phospho-AMPKα (Thr172) | 1:1,000 | Mouse | Rabbit pAB | Cell Signaling | 2535S |
| AMPKα | 1:1,000 | Mouse | Rabbit mAB | Cell Signaling | 2532S |
| Phospho-PINK1 (Ser228) | 1:1,000 | Mouse | Rabbit pAB | Thermo Fisher Scientific | PA5-105356 |
| PINK1 | 1:1,000 | Human | Rabbit mAB | Cell Signaling | 6946S |
| Phospho-PARKIN (Ser65) | 1:1,000 | Mouse | Rabbit pAB | Thermo Fisher Scientific | PA5-114616 |
| PARKIN | 1:2,000 | Mouse | Mouse mAB | Abcam | ab77924 |
| Phospho-ULK1 (Ser555) | 1:000 | Mouse | Rabbit mAB | Cell Signaling | 5869S |
| ULK1 | 1:000 | Mouse | Rabbit mAB | Cell Signaling | 8054S |
| LC3B | 1:000 | Mouse | Rabbit mAB | Cell Signaling | 3868S |
| LysoPrime Green | 1 µM | | | Dojindo | L261 |
| Mtphagy Dye | 100 nM | | | Dojindo | MT02-10 |
| BODIPY 493/503 | 20 µM | | | Thermo Fisher Scientific | D3922 |
| MitoTracker Red FM | 200nM | | | Thermo Fisher Scientific | M22425 |
| ACTB | 1:5,000 | Bovine | Mouse mAB | Sigma Life Science | A5441 |
| Beta-tubulin | 1:5,000 | Bovine | Mouse mAB | Sigma Life Science | T4026 |
| Alpha-tubulin | 1:200 | Bovine | Mouse mAB | Abcam | ab7291 |
| HRP-linked | 1:10,000 | Anti-rabbit | | Jackson ImmunoResearch | 111035003 |
| HRP-linked | 1:10,000 | Anti-mouse | | Jackson Laboratory | 115035205 |
| DyLight 405 | 1:500 | Anti-mouse | | Jackson Laboratory | 115-475-166 |
| Alexa Fluor 488 | 1:500 | Anti-mouse | | Invitrogen | A32723 |
| Alexa Fluor 594 | 1:500 | Anti-rabbit | | Invitrogen | A-11032 |
| Alexa Fluor 647 | 1:500 | Anti-biotin | | Biolegend | 405237 |

[a]Dilution used for Western blotting.
[b]Dilution used for confocal microscopy.
[c]Biotinylated antibody.
Dynamin-related protein 1 (DRP1); mitochondrial fission factor (MFF); hormone sensitive lipase (HSL); steroidogenic acute regulatory protein (STAR); cholesterol side-chain cleavage enzyme (CYP11A1); 3beta-Hydroxysteroid dehydrogenase (HSD3B); mitochondrial import receptor subunit 20 (TOM20); voltage-dependent anion-selective channel 1 (VDAC1); NF kappa B inhibitor alpha (IkBalpha); cytochrome complex (Cyt C); AMP-activated protein kinase (AMPK); PTEN-induced kinase 1 (PINK1); Parkin RBR E3 ubiquitin-protein ligase (PAKIN); Unc-51-like autophagy activating kinase (ULK1); autophagy marker Light Chain 3 (LC3); beta-actin (ACTB; loading control); beta-tubulin (TUBB; loading control); alpha-tubulin (TUBA).

point. Fold increases because of the treatment (control versus PGF2α, TNFα or PMA) were then calculated.

### Transmission electron microscopy
To observe differences in mitochondrial networks, enriched small and large luteal cells were fixed in 3% (wt/vol) paraformaldehyde and 0.2% glutaraldehyde in PBS, pH 7.4, post-fixed in 2% $OsO_4$, resin-embedded, and subject to ultra-thin sectioning for electron microscopy. Transmission electron microscopy images were captured using a FEI Tecnai G2 Spirit transmission electron microscope at the University of Nebraska Medical Center.

To evaluate the effects of PGF2α on mitochondrial morphology, enriched large luteal cells were equilibrated in fresh culture medium enriched with 1% BSA for 2 h before treatment with control media or PGF2α (100 nM) for 4 h. Following PGF2α stimulation, enriched large luteal cells were fixed and processed as described above and subject to ultra-thin sectioning for electron microscopy. Transmission electron microscopy images were captured using a FEI Tecnai G2 Spirit transmission electron microscope at the University of Nebraska Medical Center. 10–15 images (magnification: 21,000x and 35,900x) from large luteal cells were used for quantification of the mitochondrial area using ImageJ.

### Confocal microscopy
For all confocal experiments, sterile No. 1 glass coverslips (22 × 22 mm) were individually placed in each well of a six-well culture dish. Mixed luteal cells were seeded at $3 \times 10^5$ cells/well and enriched large luteal cell cultures were seeded at $2.5 \times 10^5$ cells/well.

Biotin was added to phospho-DRP1 (Ser616) and phospho-MFF (Ser146) using a commercially available kit per the manufacturer's protocol (DSB-X Biotin Protein Labeling Kit; Life Technologies Corporation) as previously described (78).

**Phosphorylation and localization of DRP1 and MFF** In brief, luteal cells were equilibrated in fresh culture medium enriched with 1% BSA for 2 h before treatment with PGF2α (100 nM) or PMA (20 nM) for 30 min. Cells were maintained at 37°C in an atmosphere of 95% humidified air and 5% $CO_2$ for 30 min before termination of the experiment.

Cells were fixed for 30 min with 200 μl 4% paraformaldehyde at 4°C and rinsed three times with PBS. To evaluate the effects of PGF2α on the phosphorylation of DRP1 and MFF, cell membranes were permeabilized with 200 μl 0.1% Triton-X in PBS-T (0.1% tween-20) at room temperature for 10 min. The permeabilized cells were then washed with PBS and blocked in 5% BSA for 24 h at 4°C. The cells were then rinsed and appropriate antibodies for co-localization (Table 1) were added to each coverslip and incubated at 4°C for an additional 24 h. Following incubation, the cells were rinsed three times with PBS to remove unbound antibody. The cells were then incubated with appropriate secondary antibodies (Table 1) at room temperature for 60 min. The cells were rinsed three times with 1 ml PBS to remove unbound antibody. After labeling with antibodies, coverslips containing labeled cells were mounted to glass microscope slides using 10 μl Fluoromount-G (Electron Microscopy Sciences). Coverslips were sealed to glass microscope slides using clear nail polish and stored at –20°C until imaging.

**Mitochondrial morphology** Luteal cell cultures were equilibrated in fresh culture medium enriched as described above and stimulated with PGF2α (100 nM) for 4 h and maintained at 37°C in an atmosphere of 95% humidified air and 5% $CO_2$. Mitotracker DeepRed (250 nM) was added to culture media 30-min before imaging. The cells were washed fresh culture media and subject to live-cell imaging.

**ROS production** We used CellROX, a weakly fluorescent cell-permeant dye that exhibits bright green photostable-fluorescence after oxidation by ROS production. In brief, luteal cell cultures were stimulated with PGF2α for 4 h and maintained at 37°C in an atmosphere of 95% humidified air and 5% $CO_2$. CellROX reagent (5 μM) was added to cultures 30 min before live-cell imaging.

**Activation of mitophagy** We used Mitophagy Dye, a weakly fluorescent mitochondrial dye that exhibits bright photostable fluorescence after fusion with lysosomes. In brief, large luteal cells were washed with PBS and placed in a fresh serum-free medium containing 100 μmol/l Mitophagy Dye. Cells were stimulated with PGF2α and maintained at 37°C in an atmosphere of 95% humidified air and 5% $CO_2$ for 6 h. To observe the co-localization of Mitophagy Dye and lysosome, the cells were incubated with Lyso dye 30 min before imaging at 37°C. The cells were washed 3× with Hanks' HEPES buffer and subject to live-cell imaging.

Images were collected using a Zeiss LSM800 confocal microscope with Airyscan equipped with a 63× oil immersion objective (1.4 N.A) and acquisition image size of 1,584 × 1,584 pixel (77.96 μm × 77.96 μm), and 1,024 × 1,024 pixel (101.31 μm × 101.31 μm). The appropriate filters were used to excite each fluorophore and emission of light was collected between 450 to 1,000 nm. Cells were randomly selected from each slide and 30–45 z-stacked (0.15 μm) images were generated from the bottom to the top of each experiment. To determine the effects of PGF2α on mean fluorescence intensity of phospho-DRP1, and phospho-MFF, images were converted to maximum intensity projections and processed utilizing ImageJ (National Institutes of Health) analysis software. Mean fluorescence intensity was determined as previously described (78). The JACoP plug-in was used in Image J software to determine the Manders' overlap coefficient for each image as previously described and transformed into percent colocalization by multiplying Manders' overlap coefficient by 100 for all colocalization experiments. Mitochondria morphology was determined using the Mitochondrial Network Analysis (MiNA) toolset package for ImageJ as described in (79).

### Mitochondria isolation

Mixed luteal cell cultures were plated in 150 × 22 mm culture dishes at $10 \times 10^6$ cells/dish. Cells were treated with culture medium alone, PGF2α (100 nM) or PMA (20 nM) for 1 h at 37°C in an atmosphere of 95% air and 5% $CO_2$. After incubation, the cells were immediately

PGF2α induces mitochondrial fission and mitophagy   Plewes et al.
https://doi.org/10.26508/lsa.202301968   vol 6 | no 7 | e202301968   **16 of 19**

rinsed with ice-cold PBS and mitochondria were isolated per the manufacturer's protocol.

## Statistical analysis

Each experiment was performed at least three times, each using cell preparations from separate cows and dates of collection. The differences in means were analyzed by one-way ANOVA followed by Tukey's multiple comparison tests to evaluate multiple responses, one-way ANOVA followed by Dunnett's post test to compare means or by $t$ tests to evaluate paired responses. Two-way ANOVA was used to evaluate repeated measures with Dunnett's posttests to compare means. All statistical analysis was performed using GraphPad Prism software from GraphPad Software, Inc. All data are presented as the means ± SEM.

# Data Availability

All data are available in the main text or the supplementary materials.

# Supplementary Information

# Acknowledgements

The authors thank Pan Zhang, Guojuan Li, and Anika Shelrud for their assistance with cell preparation. The authors also thank Janice Taylor and James Talaska at the University of Nebraska Medical Center, Advanced Microscopy Core Facility for their assistance with microscopy. The use of microscope was supported by the Center for Cellular Signaling CoBRE-P30GM106397 from the National Health Institutes. USDA National Institute of Food and Agriculture grant 2018-67012-29531 (MR Plewes). USDA National Institute of Food and Agriculture grant 2017-67015-26450 (JS Davis). U.S. Department of Veterans Affairs IK2 BX004911-01A1 (MR Plewes). U.S. Department of Veterans Affairs I01 BX004272 (JS Davis). VA Senior Research Career Scientist Award IK6BX005797 (JS Davis). National Health Institute grants R01 HD087402 (JS Davis). National Health Institute grants R01HD092263 (JS Davis). The Olson Center for Women's Health (JS Davis).

## Author Contributions

MR Plewes: conceptualization, data curation, formal analysis, funding acquisition, investigation, methodology, project administration, and writing—original draft, review, and editing.
E Przygrodzka: investigation and writing—review and editing.
CF Monaco: investigation and writing—review and editing.
AP Snider: resources and writing—review and editing.
JA Keane: data curation and writing—review and editing.
PD Burns: resources and writing—review and editing.
JR Wood: resources and writing—review and editing.
AS Cupp: resources and writing—review and editing.
JS Davis: conceptualization, resources, supervision, funding acquisition, project administration, and writing—review and editing.

## Conflict of Interest Statement

The authors declare that they have no conflict of interest.

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
