## [Reviewer comments · Life Science Alliance]

Life Science Alliance

Prostaglandin F₂ α regulates mitochondrial dynamics and mitophagy in the bovine corpus luteum

Michele Plewes, Emilia Przygodka, Corrine Monaco, Alexandria Snider, Jessica Keane, Patrick Burns, Jennifer Wood, Andrea Cupp, and John Davis

DOI: <https://doi.org/10.26508/lsa.202301968>

Corresponding author(s): Michele Plewes, University of Nebraska Medical Center and John Davis,

Review Timeline:	Submission Date:	2023-02-03
	Editorial Decision:	2023-03-07
	Revision Received:	2023-03-30
	Editorial Decision:	2023-04-14
	Revision Received:	2023-04-21
	Accepted:	2023-04-24

Transaction Report:

March 7, 2023

Re: Life Science Alliance manuscript #LSA-2023-01968

Dr. Michele R Plewes
University of Nebraska Medical Center
42nd and, Emile St
Omaha, NE 68105-1850

Dear Dr. Plewes,

Thank you for submitting your manuscript entitled "Prostaglandin F₂ α regulates mitochondrial dynamics and mitophagy in the ovarian corpus luteum" to Life Science Alliance. The manuscript was assessed by expert reviewers, whose comments are appended to this letter. We invite you to submit a revised manuscript addressing the Reviewer comments.

Thank you for this interesting contribution to Life Science Alliance. We are looking forward to receiving your revised manuscript.

Sincerely,

B. MANUSCRIPT ORGANIZATION AND FORMATTING:

Reviewer #1 (Comments to the Authors (Required)):

This manuscript by Plewes and colleagues seeks to illuminate the impact of PGF2a on mitochondria in the context of corpus luteum function and luteolysis. PGF2a is widely regarded to be luteolytic, and this body of work presents a compelling case that PGF2a treatment decreases progesterone production via alterations in mitochondrial structure/function. Detailed experiments highlight many important structural and function changes in luteal cells in response to PGF2a treatment. Methods are well described, with a large number of appropriate techniques being utilized. The combination of in vitro and in vivo studies are complimentary and provide support for the Authors' conclusions. The manuscript is unnecessarily long but otherwise well-written. The Figures are generally well presented and include images of representative experiments, graphical representation of complete data sets, and concept diagrams. Most figures represent a specific portion of the overall pathway, providing steps towards the complete story. Overall, this stepwise organization makes for a compelling presentation. Major and minor concerns are listed below.

Major Concerns:

1. The title, summary, abstract, and introduction should each clarify that this study is performed in bovine corpora lutea. A suggested title would replace "ovarian" with "bovine". In particular, the abstract does not mention the species being studied or the corpus luteum. This information should be emphasized in the abstract, especially when the publication will appear in a generalist journal.
2. The Introduction is very long and reads like a review article. The information is relevant and interesting, but it is not all required to provide context for the experiments presented in the manuscript. Much of this information can be summarized in the Introduction and/or moved to the Discussion.
3. It would be helpful if the summary diagram (Fig 8) included how the proposed pathway leads to decreased progesterone production. This concept should be clarified in the text of the Discussion as well. As written, it is not clear how the Authors believe mitochondrial changes are involved in the decline in P4 synthesis at luteal regression.
4. On Lines 93-94, the manuscript discusses "loss of progesterone and regression of the gland." Regression is most commonly defined as loss of the physical corpus luteum, while luteolysis commonly includes loss of structure and loss of function. Throughout the manuscript, luteolysis and regression are used as synonyms. If you choose to use these terms as synonyms, please clearly define "regression" with a functional definition.
5. Statistical methods are mentioned in several places. Consider consolidation into a single section, or clarify the specific experiments that each section relates to. Justification for use of non-parametric vs parametric tests is not provided. For each experiment, please provide details regarding the specific test used. This is easily done in Figure legends.
6. The Discussion does not consider limitations of the present study. In addition, the Discussion does not consider differences between bovine and human regulation of the corpus luteum that may impact translatability of the work presented.

Minor Concerns:

7. In general, background material and conclusions should be removed from the Results section. For example, Lines 169-172 and Lines 184-185 contain information that are repeated elsewhere in the manuscript. Similar examples exist throughout the Results section. It is understood that this is an issue of style, but this type of editing could reduce overall length of this very long manuscript.
8. Justification should be provided for the dose of PGF2a used in vitro. Similarly, the luteolysis-inducing dose of PGF2a is not stated on line 577. It should be clarified if this is the 2-dose treatment regimen mentioned above.
9. Progesterone assays. It is clear that tissue extracts, culture media, and plasma were assayed. Method of preparation for tissue extracts is clear, but it is not clear how progesterone from tissue extracts is determined. Specific ELISA and RIA kits are mentioned in Materials (RIA is repeated under progesterone analysis). Intra- and inter-assay CVs are provided for the RIA but not the ELISA.
10. Lines 726-783. Hypothesis/experimental design information should be removed from section headings and text. These sections should focus on methods only.
11. Line 783. It appears a word or citation is missing from the end of this sentence.
12. In general, violin plots are most useful for large data sets. For small data sets, they don't convey more information than individual data points. Please be sure that the violin plots add value and do not distract from the data presented. For example, see Fig 3G and 3H. These data may be easier to understand if presented in traditional bar graphs.
13. Citations could provide support for the statements in lines 314-316.
14. The use of a category X-axis to present changes over time is inaccurate. Consider use of a numerical X-axis and traditional

line graph.

15. Line 121. Is this sentence complete, or is something missing?

Reviewer #2 (Comments to the Authors (Required)):

The manuscript by Plewes and co-workers is an extensive, very thorough investigation of the early effects of prostaglandin F₂-alpha (PGF) on cellular events within the bovine corpus luteum involving mitochondrial fission and mitophagy. In particular, the authors provide an exhaustive, if not unwieldy, examination of PKC/ERK and AMPK events as they relate to phosphorylation of mitochondrial fission proteins, DRP1 and MFF. Additionally, the occurrence of intracellular reactive oxygen species resulting from compromised mitochondrial function is reported, as well as the promotion of PINK-Parkin mitophagy. Much of the latter work appears to be previously unreported, so it constitutes a new perspective about the actions of PGF within the bovine corpus luteum as a luteolytic mediator. The authors are to be commended for their efforts in this very detailed and extensive study, which involved both in vivo and in vitro approaches to address the proposed aims. The manuscript is fairly straightforward in its logic and progression as the investigators systematically examine first the effects of PGF on progesterone production, steroidogenic enzymes and phosphorylation of proteins critical to mitochondrial fission (DRP1 and MFF), and then pursue effects of PGF/AMPK on reactive oxygen species and mitophagy. Overall, the story presented is compelling, but also could be improved by greater focus and conciseness to the writing in some areas and some of the figures. Offered below are some suggestions to help improve the manuscript:

Lines 109-: This is the first paragraph to set up the focus of the study on mitochondrial fission and mitophagy, yet the first line of the next paragraph (Line 123) is actually the better topic sentence to begin this introduction. A suggestion would be to reconsider how these 2 paragraphs are presented, including the possibility of paring down further/consolidating the first two paragraphs of the Introduction (Lines 74- and lines 88-) to arrive at the focus of the study more quickly.

Line 112: C-terminal GTPase effector domain... this is where (GED) should be defined as the abbreviation to be used later on in the same paragraph (Line 120).

Line 123: As indicated previously, this topic sentence establishes the expectation that a brief description of mitochondrial fission AND mitophagy would follow. However, only mitophagy is described in the paragraph.

Line 158: This sentence with "animals" infers species other than, or in addition to, bovine were utilized. Avoid using the term "animals" throughout, and replace with bovine, cows, etc.

Line 156: Considering this first section is really a validation that PGF caused a decline of progesterone production without effects on steroidogenic enzymes, could it be instead relegated to supplemental information? These outcomes have been reported previously by others and are consistent with those reports.

Lines 169- and others: It seems the authors have opted in each section of the results to begin with cited work as a justification for the results that follow. This is unnecessary and should instead be incorporated into the DISCUSSION section. For this first paragraph, starting at Line 172 ("To determine the effects....") would seem to work perfectly fine for beginning this portion of the RESULTS.

Line 199: bovine large luteal cells; not "large bovine luteal cells"

Lines 208-212: Although much of these results are part of the supplemental information, did the authors also evaluate the phosphorylation of Ser637 in the experiments? What was the outcome of this? Does it have physiological relevance?

Lines 222-224: In this section it is evident that PGF increased phosphorylation of Ser637, as alluded to in the inquiry above. What does this mean biologically-speaking? And what about the effects of PMA and TNF?

Line 243: Here, the authors now describe the effect of PMA on phosphorylation of Ser637, which seems to make the effect of TNF on DRP1 (Ser637) relevant. A suggestion for the above sections would be to somehow consolidate and consistently report the results so that the "Effects of hormones on phosphorylation of DRP1...", "Temporal effects of luteolytic hormones on the phosphorylation of DRP1", and "Effects of PKC and MAPK signaling...." don't appear as disjointed/disconnected.

Line 296: Suggest revising to write, "The results thus far support the hypothesis that PGF acutely influences mitochondrial dynamics through phosphorylation of DRP1 and MFF. In light of this, using confocal microscopy, we set out...."

Line 338: Should this be reference 45, not 44?

Results: General---It might be helpful to reorganize the reporting of the results in a similar manner to that of the methodology. That is, report all of the in vivo results first, followed by the in vitro results, and then delineate tissue-level phenomena (e.g.,

morphology) followed by cell specific and molecular events (e.g., large luteal cells and then phosphorylation of molecules within). The back-and-forth that occurs is distracting.

Figures 2 and 3: General---is a full 24 hr time course set of experiments necessary to report in the results considering that much of the initial in vivo work is based upon the first 4 hrs following PGF? Some of these time course results could be reported as supplemental information instead to simplify the figures. They seem more like validation of methodology rather than central to the point of phosphorylation, or lack of phosphorylation, of DRP1 and MFF in response to the various molecular manipulations.

In summary, although the authors have provided considerable and noteworthy experimental evidence in this study, there are some issues concerning the organization and overall presentation of the work that should be addressed to make it more logical and informative for the reader.

LSA-2023-01968

**Response:** We thank the reviewer for the positive comments about our manuscript and the
suggestions for improvement.

Major Concerns:

1. The title, summary, abstract, and introduction should each clarify that this study is performed
in bovine corpora lutea. A suggested title would replace "ovarian" with "bovine". In particular, the
abstract does not mention the species being studied or the corpus luteum. This information
should be emphasized in the abstract, especially when the publication will appear in a generalist
journal.

**Response:** As suggested, bovine corpus luteum is mentioned in the title, abstract and
Introduction.

2. The Introduction is very long and reads like a review article. The information is relevant and
interesting, but it is not all required to provide context for the experiments presented in the
manuscript. Much of this information can be summarized in the Introduction and/or moved to the
Discussion.

**Response:** Thank you for the suggestion. We have shortened the introduction as requested.

3. It would be helpful if the summary diagram (Fig 8) included how the proposed pathway leads
to decreased progesterone production. This concept should be clarified in the text of the
Discussion as well. As written, it is not clear how the Authors believe mitochondrial changes are
involved in the decline in P4 synthesis at luteal regression.

**Response:** We have not modified the summary figure because we feel that adding an
additional layer to the already detailed illustration of mitochondrial responses would be
complicated and potentially confusing. We have added additional text to Discussion as
suggested. Mitochondrial function is necessary for conversion of cholesterol to pregnenolone,
the penultimate step before progesterone synthesis. Moreover, previous studies from our
laboratory (1) and others (2) have demonstrated that mitochondrial fusion is a key process
necessary for optimal steroidogenesis.

4. On Lines 93-94, the manuscript discusses "loss of progesterone and regression of the gland."
Regression is most commonly defined as loss of the physical corpus luteum, while luteolysis
commonly includes loss of structure and loss of function. Throughout the manuscript, luteolysis
and regression are used as synonyms. If you choose to use these terms as synonyms, please
clearly define "regression" with a functional definition.

**Response:** Thank you, we have provided clarity: ...at the onset of luteolysis, there is a
precipitous decline in concentrations of progesterone in serum (functional regression) followed
by loss of luteal weight (structural regression). We have also uniformly used the term luteolysis
in the text instead of regression.

5. Statistical methods are mentioned in several places. Consider consolidation into a single

section, or clarify the specific experiments that each section relates to. Justification for use of
non-parametric vs parametric tests is not provided. For each experiment, please provide details
regarding the specific test used. This is easily done in Figure legends.

**Response:** Thank you for the suggestion. In the revised manuscript, all statistical analysis has
been performed using parametric testing. The statistical methods have been consolidated into a
single section within the methods section (Line 789-794). Moreover, for each experiment, we
provided details regarding the specific test used (Figure captions) as suggested.

6. The Discussion does not consider limitations of the present study. In addition, the Discussion
does not consider differences between bovine and human regulation of the corpus luteum that
may impact translatability of the work presented.

**Response:** We have added a section in the Discussion on the limitations of the study (Lines
461-466). We have also added a brief section about bovine and human regulation of the corpus
luteum (Lines 484-491).

Minor Concerns:

7. In general, background material and conclusions should be removed from the Results
section. For example, Lines 169-172 and Lines 184-185 contain information that are repeated
elsewhere in the manuscript. Similar examples exist throughout the Results section. It is
understood that this is an issue of style, but this type of editing could reduce overall length of
this very long manuscript.

**Response:** We have made edits as suggested to reduce repetition and the length of the
manuscript.

8. Justification should be provided for the dose of PGF2a used in vitro. Similarly, the luteolysis-
inducing dose of PGF2a is not stated on line 577. It should be clarified if this is the 2-dose
treatment regimen mentioned above.

**Response:** The use of 100 nM PGF2a is based on a previously published study (3). The
following study established a dose-response curve (100 nM-100 μ M). The authors report higher
concentrations of PGF2a (1 μ M) may lead to bleed over and activation of other PG receptors in
mixed bovine luteal cells. Therefore, 100 nM PGF2a was used to delineate the effects of
PTGFR signaling on mitochondrial dynamics and mitophagy in bovine large luteal cells.

For *in vivo* PGF2a administration, we have provided clarity... Cows were synchronized using
two intramuscular injections of PGF2 α (25mg; Lutalyse®, Zoetis Inc., Kalamazoo Michigan, MI)
11 days apart. At mid-cycle (days 9-10), cows were treated with an intra-muscular injection of
saline (n = 3) or PGF2 α (25mg; n = 9).

9. Progesterone assays. It is clear that tissue extracts, culture media, and plasma were
assayed. Method of preparation for tissue extracts is clear, but it is not clear how progesterone
from tissue extracts is determined. Specific ELISA and RIA kits are mentioned in Materials (RIA
is repeated under progesterone analysis). Intra- and inter-assay CVs are provided for the RIA
but not the ELISA.

**Response:** Thank you for bringing it to our attention. We have included the Intraassay CV for
the ELISA assays that was used in the following study and specified which steroid assay was
used for each experiment (Lines 601-602).

10. Lines 726-783. Hypothesis/experimental design information should be removed from section
headings and text. These sections should focus on methods only.

**Response:** The text has been modified as requested.

11. Line 783. It appears a word or citation is missing from the end of this sentence.

**Response:** Thank you. The reference for the statement has been added.

12. In general, violin plots are most useful for large data sets. For small data sets, they don't
convey more information than individual data points. Please be sure that the violin plots add
value and do not distract from the data presented. For example, see Fig 3G and 3H. These data
may be easier to understand if presented in traditional bar graphs.

**Response:** We have reviewed our use of the violin plots and have replaced some with
traditional bar graphs, as appropriate (Figures 1B, 3D, 4D, 4G-4H, 5B-5E, 5G, 6C, 6E-6F, 7G-
7H).

13. Citations could provide support for the statements in lines 314-316.

**Response:** References for the statement have been added (Line 302-304, reference 47).

14. The use of a category X-axis to present changes over time is inaccurate. Consider use of a
numerical X-axis and traditional line graph.

**Response:** Figures representing changes over time have been modified as requested to
traditional line graphs (Figures 1A, 1C-1F, 2B-2D, 4B, 7B-7D).

15. Line 121. Is this sentence complete, or is something missing?

**Response:** Thank you, the text in this section has been modified as suggested to condense
the introduction.

**Reviewer #2**

Overall, the story presented is compelling, but also could be improved by greater focus and
conciseness to the writing in some areas and some of the figures. Offered below are some
suggestions to help improve the manuscript:

**Response:** We thank the reviewer for the positive comments about our manuscript and the
suggestions for improvement.

Lines 109-: This is the first paragraph to set up the focus of the study on mitochondrial fission
and mitophagy, yet the first line of the next paragraph (Line 123) is actually the better topic
sentence to begin this introduction. A suggestion would be to reconsider how these 2
paragraphs are presented, including the possibility of paring down further/consolidating the first
two paragraphs of the Introduction (Lines 74- and lines 88-) to arrive at the focus of the study
more quickly.

**Response:** We thank the reviewer for the suggestion to compile the first two paragraphs. This
has been edited as suggested.

Line 112: C-terminal GTPase effector domain... this is where (GED) should be defined as the
abbreviation to be used later on in the same paragraph (Line 120).

**Response:** Thank you. This has been moved as suggested.

Line 123: As indicated previously, this topic sentence establishes the expectation that a brief
description of mitochondrial fission AND mitophagy would follow. However, only mitophagy is
described in the paragraph.

**Response:** Thank you. The following sentence has been modified as suggested.

Line 158: This sentence with "animals" infers species other than, or in addition to, bovine were
utilized. Avoid using the term "animals" throughout, and replace with bovine, cows, etc.

**Response:** Thank you. This has been edited in the revised manuscript as suggested.

Line 156: Considering this first section is really a validation that PGF caused a decline of
progesterone production without effects on steroidogenic enzymes, could it be instead relegated
to supplemental information? These outcomes have been reported previously by others and are
consistent with those reports.

**Response:** Thank you. The effects of PGF2a on the expression of steroidogenic enzymes has
been relegated to supporting information Figure 1 as suggested.

Lines 169- and others: It seems the authors have opted in each section of the results to begin
with cited work as a justification for the results that follow. This is unnecessary and should
instead be incorporated into the DISCUSSION section. For this first paragraph, starting at Line
172 ("To determine the effects...") would seem to work perfectly fine for beginning this portion
of the
RESULTS.

**Response:** Thank you for the suggestion. We have made the appropriate modifications in the
results section.

Line 199: bovine large luteal cells; not "large bovine luteal cells"

**Response:** This has been corrected in the revised manuscript.

Lines 208-212: Although much of these results are part of the supplemental information, did the
authors also evaluate the phosphorylation of Ser637 in the experiments? What was the
outcome of this? Does it have physiological relevance?

**Response:** In response to luteolytic stimuli we observe ~10 fold elevations in phosphorylation
of DRP1Ser616 and a modest increase (~2 fold) in phosphorylation of DRP1 Ser637. The
mechanism causing the phosphorylation of DRP-Ser637 in response to luteolytic stimuli is not
understood at present, but could result from inhibition of a phosphatase, or negative feedback
from the very robust activation of signals as a result of PGF. Regardless, the ratio of pDRP1-
Ser616 to pDRP1-Ser637 DRP1 is about 5 and appears to be sufficient to induce mitochondrial
fragmentation. In contrast, the luteotropic hormone, LH, increases pDRP1-Ser637 (inhibits
DRP1 GTPase activity) and reduces pDRP1-Ser616 (reduced recruitment to mitochondria),
resulting in mitochondrial elongation (1). This leads us to believe that modest elevation in
pDRP1-Ser637 does not prevent DRP1 recruitment to mitochondrial and stimulation of
fragmentation.

Lines 222-224: In this section it is evident that PGF increased phosphorylation of Ser637, as
alluded to in the inquiry above. What does this mean biologically-speaking? And what about the
effects of PMA and TNF?

**Response:** As described above, we are uncertain about the biological relevance of PGF2a
induced phosphorylation of DRP1 at Ser637. Treatment with PKC activator, PMA, had no
influence on phosphorylation of DRP1 at Ser637 (Figure 4F). Moreover, since DRP1 is recruited
to mitochondrial and fission occurs following treatment with PGF2a, the biological meaning of
increased pDRP1-Ser637 is uncertain. In the initial submission, we did not quantify the
phosphorylation of DRP1 at Ser637 in response to TNFa. Therefore, to focus our study, we
have removed the representative data panel to pDRP1-Ser637 (Supporting Information Figure
3A).

Line 243: Here, the authors now describe the effect of PMA on phosphorylation of Ser637,
which seems to make the effect of TNF on DPR1 (Ser637) relevant. A suggestion for the above
sections would be to some how consolidate and consistently report the results so that the
"Effects of hormones on phosphorylation of DRP1...", "Temporal effects of luteolytic hormones
on the phosphorylation of DRP1", and "Effects of PKC and MAPK signaling...." don't appear as
disjointed/disconnected.

**Response:** Thank you. The following section has been consolidated as suggested using the
heading, "Effects of hormones on phosphorylation of DRP1 in bovine large luteal cells."

Line 296: Suggest revising to write, "The results thus far support the hypothesis that PGF
acutely influences mitochondrial dynamics through phosphorylation of DPR1 and MFF. In light
of this, using confocal microscopy, we set out...."

**Response:** Thank you, modified as suggested.

Line 338: Should this be reference 45, not 44?

**Response:** Thank you, this has been modified to include both references (Manuscript and
subsequent data in brief).

Results: General---It might be helpful to reorganize the reporting of the results in a similar
manner to that of the methodology. That is, report all of the in vivo results first, followed by the in
vitro results, and then delineate tissue-level phenomena (e.g., morphology) followed by cell
specific and molecular events (e.g., large luteal cells and then phosphorylation of molecules
within). The back-and-forth that occurs is distracting.

**Response:** Thank you, this has been modified as suggested. The revised manuscript presents
*in vivo* results first followed by *in vitro*, cell specific results.

Figures 2 and 3: General---is a full 24 hr time course set of experiments necessary to report in
the results considering that much of the initial in vivo work is based upon the first 4 hrs following
PGF? Some of these time course results could be reported as supplemental information instead
to simplify the figures. They seem more like validation of methodology rather than central to the
point of phosphorylation, or lack of phosphorylation, of DRP1 and MFF in response to the
various molecular manipulations.

**Response:** Thank you, the following has been modified as suggested.

In summary, although the authors have provided considerable and noteworthy experimental
evidence in this study, there are some issues concerning the organization and overall
presentation of the work that should be addressed to make it more logical and informative for
the reader.

- 1. Plewes, M. R., Hou, X., Talbott, H. A., Zhang, P., Wood, J. R., Cupp, A. S., and Davis, J.
S. (2020) Luteinizing hormone regulates the phosphorylation and localization of the
mitochondrial effector dynamin-related protein-1 (DRP1) and steroidogenesis in the
bovine corpus luteum. *The FASEB Journal* **34**, 5299-5316
- 2. Duarte, A., Poderoso, C., Cooke, M., Soria, G., Cornejo Maciel, F., Gottifredi, V., and
Podestá, E. J. (2012) Mitochondrial fusion is essential for steroid biosynthesis.
- 3. Davis, J. S., Weakland, L. L., Weiland, D. A., Farese, R. V., and West, L. A. (1987)
Prostaglandin F2 alpha stimulates phosphatidylinositol 4, 5-bisphosphate hydrolysis and
mobilizes intracellular Ca²⁺ in bovine luteal cells. *Proceedings of the National Academy
of Sciences* **84**, 3728-3732

April 14, 2023

RE: Life Science Alliance Manuscript #LSA-2023-01968R

Dr. Michele R Plewes
University of Nebraska Medical Center
42nd and, Emile St
Omaha, NE 68105-1850

Dear Dr. Plewes,

Thank you for submitting your revised manuscript entitled "Prostaglandin F₂ α regulates mitochondrial dynamics and mitophagy in the bovine corpus luteum". We would be happy to publish your paper in Life Science Alliance pending final revisions necessary to meet our formatting guidelines.

- please upload your supplementary figures as single files and rename them as supplementary figures rather than supporting information figures; please adjust the figure callouts in the text and in the legend accordingly (Figure S1A, S1B, etc.)
- you can remove your Figure 8 figure legend and Figure 8 callout, as this will be designated as a graphical abstract rather than a figure
- please incorporate the Conclusion section into your main Discussion section
- please upload your tables as editable doc or excel files
- please double-check your figure callouts for Figure 2; you have a callout for the panel F, but it seems like this should be the panel E
- please add a figure callout for Figure S1B, Figure S3A-C and Figure S5

Figure Check:

- please provide original blots as Source Data files for Figure 2, Figure 3A and Figure 4E

A. FINAL FILES:

B. MANUSCRIPT ORGANIZATION AND FORMATTING:

Sincerely,

Reviewer #2 (Comments to the Authors (Required)):

The resubmitted manuscript by Plewes and co-workers is a considerable improvement to the extensive study previously reported. In brief, the authors provide evidence for early effects of prostaglandin F₂-alpha (PGF) on cellular events within the bovine corpus luteum (CL) involving mitochondrial fission and mitophagy. The examination of PKC/ERK and AMPK events as they relate to phosphorylation of mitochondrial fission proteins, DRP1 and MFF, is important new information for the CL literature. Similarly, intracellular reactive oxygen species resulting from compromised mitochondrial function, as well as the promotion of PINK-Parkin mitophagy, constitutes a new perspective about the actions of PGF within the bovine CL as a luteolytic mediator. The authors have adequately addressed previously-provided reviewer comments, including the consolidation or reorganization of text throughout to focus the work and to make for a more logical presentation of the findings. The flow of progression is consistently in vivo work followed by in vitro work, which is appreciated. Overall, the story remains compelling, but the revised manuscript is much improved by the greater focus and conciseness to the writing and the figures.

April 24, 2023

RE: Life Science Alliance Manuscript #LSA-2023-01968RR

Dr. Michele R Plewes
University of Nebraska Medical Center
42nd and, Emile St
Omaha, NE 68105-1850

Dear Dr. Plewes,

Thank you for submitting your Research Article entitled "Prostaglandin F₂ α regulates mitochondrial dynamics and mitophagy in the bovine corpus luteum". It is a pleasure to let you know that your manuscript is now accepted for publication in Life Science Alliance. Congratulations on this interesting work.

DISTRIBUTION OF MATERIALS:

Again, congratulations on a very nice paper. I hope you found the review process to be constructive and are pleased with how the manuscript was handled editorially. We look forward to future exciting submissions from your lab.

Sincerely,
